# Regulation of mitochondria-dynactin interaction and mitochondrial retrograde transport in axons

Catherine M Drerup[1,2]*, Amy L Herbert[3], Kelly R Monk[3], Alex V Nechiporuk[1]*

[1]Department of Cell, Developmental and Cancer Biology, Oregon Health & Science University, Portland, United States; [2]National Institute of Child Health and Human Development, National Institutes of Health, Bethesda, United States; [3]Department of Developmental Biology, Washington University School of Medicine, St. Louis, United States

**Abstract** Mitochondrial transport in axons is critical for neural circuit health and function. While several proteins have been found that modulate bidirectional mitochondrial motility, factors that regulate *unidirectional* mitochondrial transport have been harder to identify. In a genetic screen, we found a zebrafish strain in which mitochondria fail to attach to the dynein retrograde motor. This strain carries a loss-of-function mutation in *actr10*, a member of the dynein-associated complex dynactin. The abnormal axon morphology and mitochondrial retrograde transport defects observed in *actr10* mutants are distinct from dynein and dynactin mutant axonal phenotypes. In addition, Actr10 lacking the dynactin binding domain maintains its ability to bind mitochondria, arguing for a role for Actr10 in dynactin-mitochondria interaction. Finally, genetic interaction studies implicated Drp1 as a partner in Actr10-dependent mitochondrial retrograde transport. Together, this work identifies Actr10 as a factor necessary for dynactin-mitochondria interaction, enhancing our understanding of how mitochondria properly localize in axons.

*For correspondence: katie. drerup@nih.gov (CMD); nechipor@ohsu.edu (AVN)

**Competing interests:** The authors declare that no competing interests exist.

## Introduction

Mitochondrial transport in axons is critical for the formation and function of the nervous system. This organelle generates the ATP necessary for energy demanding functions in all cells, but neurons are especially reliant on mitochondria to maintain their electrically polarized state. After depolarization, ATP-dependent ion pumps are employed to repolarize the cell and prepare it for another action potential. It is estimated that at rest alone, neurons use 4.7 billion molecules of ATP per second (*Zhu et al., 2012*). Because of this large ATP requirement, mitochondria need to be properly localized to regions of high ion influx, such as at synapses (reviewed in *Schwarz, 2013*). In addition to their critical role in cellular metabolism, mitochondria also regulate local calcium ion levels (*Werth and Thayer, 1994*). Calcium efflux from intracellular stores mediates synaptic activity. During inactivity, this ion needs to be contained in intracellular compartments as high cytoplasmic calcium levels correlate with axonal degeneration (*Avery et al., 2012*; *Vargas et al., 2015*; *Yang et al., 2013*). The proper localization of mitochondria to sites of high ATP consumption and calcium ion flux requires active transport.

In addition to maintaining axon health, mitochondrial motility is also necessary to maintain mitochondrial health and function. Mitochondria undergo fission-fusion dynamics that facilitate both the replenishment of proteins in this organelle and the maintenance of mitochondrial DNA quantity and integrity (reviewed in *Scheibye-Knudsen et al., 2015*). Interrupted mitochondrial fission or fusion has been linked to loss of mitochondrial DNA, loss of oxidative potential, and mitophagy. Axonal

transport of mitochondria is tightly linked to mitochondrial dynamics as mitochondrial fusion requires the coalescence of mitochondria and fission requires the active separation of dividing organelles. The relationship between mitochondrial dynamics and transport is also apparent at the molecular level: Mitofusin, an essential protein for mitochondrial fusion, participates in the anterograde transport of this organelle as well (*Misko et al., 2010*). Furthermore, manipulation of the dynamin-like protein Drp1, necessary for mitochondrial fission, impacts the localization of mitochondria (*Smirnova et al., 2001*; *Varadi et al., 2004*). The mechanistic bases for these relationships are still largely unclear.

Elegant work in *Drosophila*, *C. elegans*, and cultured neurons has begun to elucidate the mechanisms of mitochondrial axonal transport. The primary anterograde mitochondrial motor, Kinesin-1, attaches to mitochondria via the proteins Miro (RhoT1/2; *Guo et al., 2005*) and Milton (TRAK1/2; *Glater et al., 2006*; *Stowers et al., 2002*). Miro contains calcium sensitive EF hands that, when exposed to high levels of this ion, change their confirmation, resulting in the uncoupling of the Kinesin-1 motor from microtubules (*Saotome et al., 2008*; *Wang and Schwarz, 2009*). This mechanism, in conjunction with mitochondrial tethering factors (*Kang et al., 2008*), allows mitochondrial congregation at sites of high synaptic activity. The Miro/Milton transport machinery is not specific for anterograde transport, however, as loss of either protein impacts anterograde and retrograde mitochondrial movement (*Guo et al., 2005*; *Saotome et al., 2008*). Therefore, how mitochondria are specifically moved in a unidirectional manner by either the Kinesin or Dynein motor protein complex is not well understood.

Retrograde mitochondrial transport is known to depend on the cytoplasmic dynein complex (*Pilling et al., 2006*; *Schnapp and Reese, 1989*). The core dynein motor (reviewed in *Holzbaur and Vallee, 1994*) is oftentimes associated with dynactin, itself a multi-protein complex, during retrograde axonal transport. Dynactin, specifically its p150 subunit, has been shown to facilitate dynein processivity and also serves as an anchor for dynein at microtubule plus ends, facilitating cargo loading (*Lloyd et al., 2012*; *Moughamian and Holzbaur, 2012*; *Schroer, 2004*). The dynactin accessory complex is attached to dynein through interaction of p150 with the tails of the dynein intermediate chains (*Vaughan and Vallee, 1995*). In addition to p150, dynactin contains two actin-related proteins, Arp1 and Actr10 (also known as Arp11; *Eckley et al., 1999*; *Eckley and Schroer, 2003*). Together with p25, p62 and p27, Actr10 is a part of the dynactin pointed end complex which is predicted to be in an ideal location for cargo binding (*Yeh et al., 2012*). Structural work suggests that one key function for Actr10 is capping the actin-like Arp1 filament, to regulate filament length and facilitate attachment of other pointed end proteins (*Urnavicius et al., 2015*); however, this does not preclude an additional role for Actr10 in cargo attachment to the dynein complex, a function which has not been explored.

In a forward genetic screen for mediators of retrograde axonal transport, we identified a strain with a loss-of-function mutation in Actr10. $actr10^{nl15}$ mutants (hereafter referred to as *actr10*) display axon terminal swellings in the central and peripheral nervous systems indicative of retrograde transport abnormalities. Analysis of cargo localization and movement in the *actr10* mutant revealed clustering of mitochondria, but not other cargos analyzed, at microtubule plus ends due to failed retrograde mitochondrial movement. This phenotype was vastly different from loss of either Dynein heavy chain or p150, potentially indicating a unique function for Actr10 in mitochondrial retrograde movement. Furthermore, we demonstrated that abnormal mitochondrial movement in *actr10* mutants is due to failed attachment of mitochondria to the dynein-dynactin complex in the absence of Actr10. Importantly, Actr10 engineered to lack the dynactin binding domain maintains mitochondrial interaction, hinting at a specific role for Actr10 in mediating dynactin-mitochondria interaction. Finally, we provide biochemical and genetic evidence that Drp1, a Dynamin-related protein previously implicated in microtubule minus end-directed mitochondrial movement (*Smirnova et al., 1998*), partners with Actr10 in mitochondrial retrograde transport. Together our data support a model in which Actr10 functions to scaffold mitochondria to the dynein-dynactin complex for retrograde transport in axons.

## Results

### Mitochondria accumulate in *actr10* mutant axon terminals due to failed retrograde transport

We used the zebrafish posterior lateral line (pLL) system to identify novel mediators of retrograde cargo transport in axons (*Drerup and Nechiporuk, 2013*). pLL axons develop early (axon extension is complete by 2 days post-fertilization (dpf) and synapse formation occurs by 4 dpf), are superficially localized and are largely planar, making them an ideal system for *in vivo* observations and manipulations (*Figure 1A*; *Ghysen and Dambly-Chaudière, 2004*; *Metcalfe, 1985*; *Metcalfe et al., 1985*). Using an ENU-based forward genetic screen, we isolated a larval-lethal, recessive mutant strain with large swellings in pLL axon terminals (*Figure 1B*). RNAmapper analysis (*Miller et al., 2013*) identified the causal mutation as a single nucleotide change (T to G) in the start codon of *actr10*. We confirmed *actr10* as the affected gene using RNA-mediated rescue of axon terminal swellings and by performing TALEN- (Transcription Activator Like Effector Nuclease) mediated disruption of the first exon of *actr10*, which phenocopied the mutant axon terminal swellings (*Figure 1C–F*).

Axonal swellings, like those observed in *actr10* mutants, can arise due to a number of intracellular abnormalities, including disruptions in retrograde cargo transport (*Drerup and Nechiporuk, 2013*;

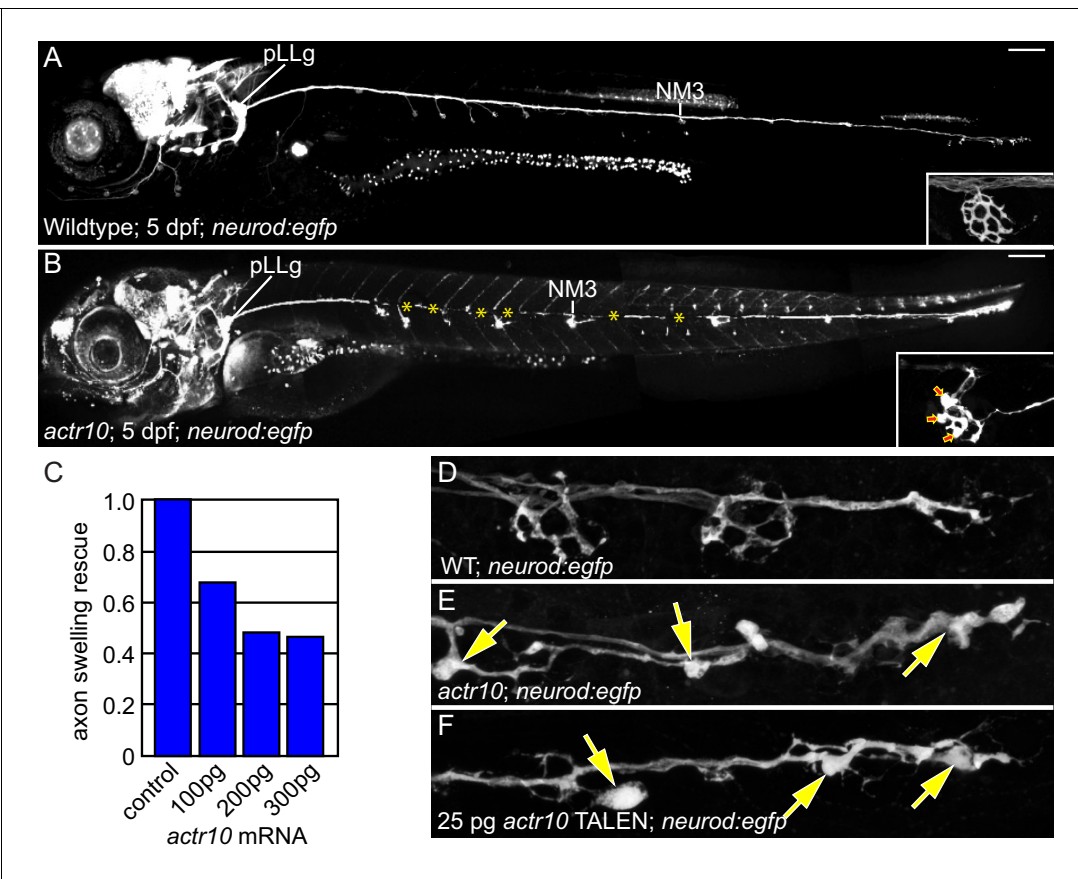

**Figure 1.** *actr10* mutants have swollen axon terminals. (**A**) Wild-type larva (pigment free) at 4 dpf carrying the *neurod:egfp* BAC transgene. By 4 dpf, pLL axons are fully extended and functional synapses have formed with hair cells of primary neuromasts. (**B**) *actr10* mutant axons fully extend but terminals display large swellings. Posterior lateral line ganglion (pLLg) and innervation of the third neuromast (NM3) are indicated. Insets in (**A**) and (**B**) show magnified NM3 axon terminals. Arrows in inset point to swellings. Asterisks label areas of the pLL nerve obscured by pigment in the mutant. (**C**) The mutant axon terminal swelling phenotype can be suppressed by exogenous expression of mRFP-Actr10 in a dose-dependent manner. Proportion of mutants with axon terminal swellings is depicted. (**D–F**) Injection of TALENS targeting exon 1 of the *actr10* genomic locus phenocopies the *actr10* mutant axon terminal swelling phenotype in F0 injected larvae (arrows). Axons are labeled by the *neurod:egfp* BAC transgene (white). Scale bars in A, B = 100 μm.

*Martin et al., 1999*). To determine if a particular cargo was accumulating in *actr10* mutant axon terminals, indicative of impaired retrograde movement, we performed immunolabeling on mutants and wildtype siblings at 4 dpf with antibodies against various axonal cargos. This revealed an accumulation of Cytochrome c, a mitochondrial protein, in *actr10* axon terminals (*Figure 2A,B,E*). To confirm mitochondrial accumulation in mutant axon terminals, mitochondria were labeled in neurons with TagRFP by zygotic injection of a plasmid containing a mitochondrial targeting sequence tagged with this fluorescent protein (*5kbneurod:mito-TagRFP*). *actr10* mutants displayed mitochondrial accumulation in pLL axon terminals (*Figure 2C,D*). Other known pLL axon retrograde cargos, including those labeled by Lamp1 (Lysosome associated membrane protein-1; late endosome/lysosome marker) and phosphorylated c-Jun N-Terminal Kinase (pJNK) did not accumulate (*Figure 2F–K*; *Drerup and Nechiporuk, 2013*). Expression of wildtype Actr10 using zygotic injection of *in vitro*

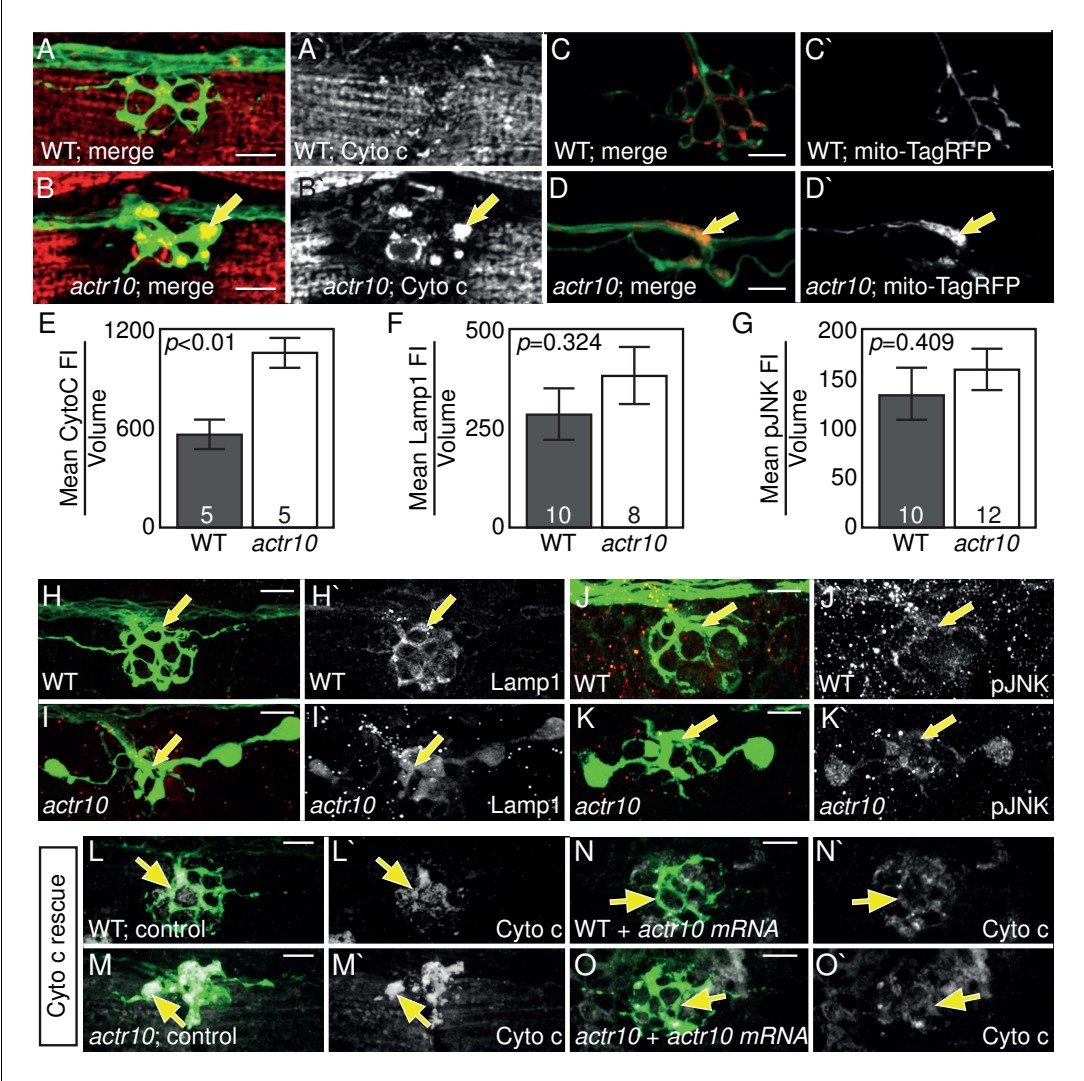

**Figure 2.** Loss of Actr10 causes mitochondrial accumulation in swollen axon terminals. (A,B) At 4 dpf, swollen axon terminals in *actr10* mutants (B) have high levels of Cytochrome c immunolabeling (arrow; NM3 shown) relative to a wildtype sibling (A). (C,D) TagRFP-labeled mitochondria accumulate in *actr10* mutant (arrow), but not wildtype, axon terminals at 4 dpf. (E) Mean fluorescence intensity (background subtracted) of Cytochrome C in axon terminals of A and B. (F,G) Mean fluorescence intensity (background subtracted) in axon terminals of H-K show comparable levels of Lamp1 and pJNK fluorescence intensity between wildtype and *actr10* mutant terminals (ANOVA; mean ± SEM shown). (H–K) Lamp1 and pJNK immunolabeling in NM3 axon terminals (arrows). Lamp1 and pJNK are in red in (H–K) and white in H'-K'. (L–O) Mitochondrial accumulation in mutant axon terminals, assayed using Cytochrome c immunolabeling (white), can be suppressed by mRNA-mediated expression of mRFP-Actr10. Arrows point to axon terminal regions for comparison. Scale bars = 10 μm.

synthesized mRNA rescued mitochondrial accumulation in *actr10* mutants (*Figure 2L–O*), confirming that Actr10 is necessary for proper mitochondrial positioning.

As our immunolabeling experiments were not exhaustive for every possible cargo in axons, we undertook transmission electron microscopy (TEM) analyses of pLL axon terminals to determine if mitochondria are the predominant cargo mislocalized in *actr10* mutants. This experiment demonstrated mitochondria are highly enriched in the axon terminal swellings of *actr10* mutants compared to other intracellular structures (*Figure 3A,B*; n = 3 wildtypes, n = 4 mutants). Together, our immunolabeling and TEM analyses showing mitochondrial accumulation in axon terminals could indicate interrupted retrograde mitochondrial transport in *actr10* mutants.

To determine if mitochondrial retrograde transport was disrupted in *actr10* mutants, mitochondria in single pLL axons were tagged with TagRFP using *5kbneurod:mito-TagRFP* plasmid injection and organelle movement was visualized by confocal microscopy (*Figure 4A,B*; *Videos 1* and *2*). Kymograph analysis demonstrated a 37% and 39% reduction in the distance moved by mitochondria in the anterograde and retrograde directions respectively in *actr10* mutants but no change in mitochondrial transport velocity (*Figure 4C–F*). Despite the decrease in anterograde transport distance, the frequency of anterograde mitochondrial transport was unchanged in *actr10* mutants. We also analyzed reversal frequency to determine if there was a defect in directional persistence of mitochondrial transport in *actr10* mutants but found no difference in the proportion of mitochondria that reversed direction during our imaging sessions (wildtype: 0.011 ± 0.008; *actr10* mutants: 0.000 ± 0.009; ANOVA; *p*=0.4003). In contrast, both the frequency of retrograde mitochondrial transport and the proportion of mitochondria moving in the retrograde direction were dramatically decreased in *actr10* mutants (*Figure 4G,H*). As velocity of movement was largely normal, these data suggest a role for Actr10 in the attachment of mitochondria to the retrograde motor for transport.

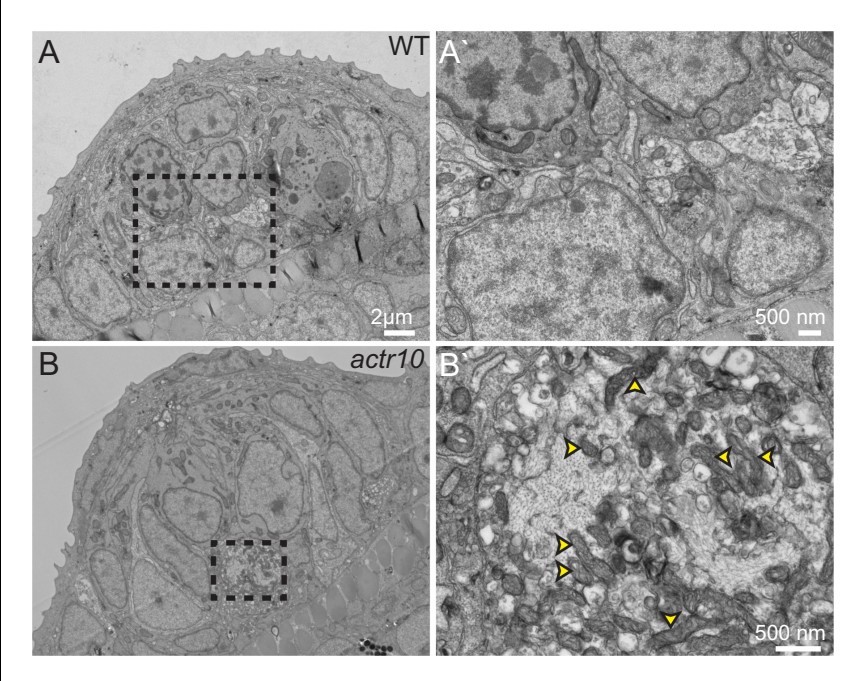

**Figure 3.** TEM analysis of axon terminal swellings reveal mitochondrial accumulation. (**A**) Wild-type axon terminals innervating lateral line neuromasts showed no swellings (N = 3). (**B**) All neuromasts assayed in *actr10* mutants had mitochondrial laden axon terminal swellings (N = 4). A' and B' are higher magnification views of the areas outlined in A and B. Arrowheads in B' point to mitochondria.

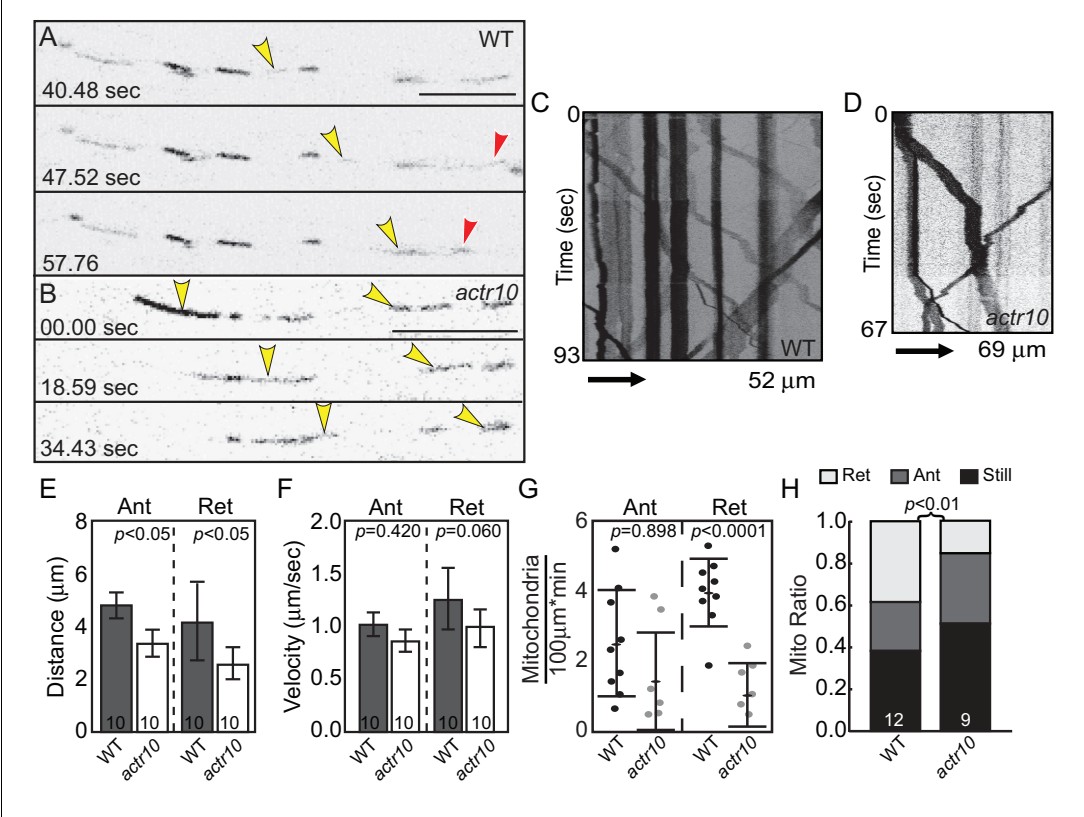

**Figure 4.** Mitochondrial retrograde transport is specifically disrupted in *actr10* mutants. (**A,B**) Stills from *Videos 1* and *2* of mitochondrial transport in single pLL axons of a wildtype (**A**) and *actr10* mutant (**B**). Yellow and red arrowheads point to mitochondria moving in the anterograde and retrograde directions respectively. (**C,D**) Kymograph analyses of mitochondrial transport in wildtype (**C**) and *actr10* mutants (**D**). (**E,F**) Distance and velocity of mitochondrial transport in *actr10* mutants (Ant-anterograde; Ret-retrograde; ANOVA). (**G**) The number of mitochondria moving in the retrograde direction is significantly reduced in *actr10* mutants (ANOVA; $p<0001$). (**H**) The proportion of mitochondria moving in the retrograde, but not anterograde, direction is reduced in *actr10* mutants (ANOVA with post-hoc contrasts; $p<0.01$). Scale bars = 10 µm. Error bars represent mean ± SEM. Number of larvae assayed (biological replicates) is indicated on graphs.

## Actr10 functions autonomously in neurons to regulate axon morphology and mitochondrial localization

*actr10* is ubiquitously expressed, albeit enriched in the nervous system during larval stages (*Figure 5A–C*). As neuronal activity and the presence of growth factors can modulate mitochondrial transport (*Chada and Hollenbeck, 2004*; *Chen and Sheng, 2013*), it was possible that changes in the axonal environment in *actr10* mutants could alter mitochondrial movement. This led us to ask if Actr10 functions autonomously in axons to regulate mitochondrial transport. To test for a neuron-specific function for Actr10 in mitochondrial localization, we expressed monomeric red fluorescent protein (mRFP) tagged Actr10 in individual pLL neurons using zygotic injection of a *5kbneurod: mRFP-actr10* DNA plasmid and assessed axon terminals. While mRFP-Actr10 expression in wild-type axons had no apparent effects (*Figure 5D*; n = 6 in 3 biological replicates), it rescued axon terminal morphology and mitochondrial localization in all mutants analyzed (*Figure 5E–G*; n = 5 mutants in 3 biological replicates). These experiments confirmed a neuron-specific function for Actr10 in mitochondrial localization in axons.

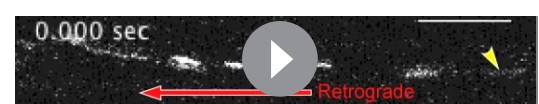

**Video 1.** Mitochondrial transport in a 4 dpf wildtype larva, related to *Figure 4*. Yellow arrowheads indicate mitochondria moving in the anterograde direction. Red arrowhead marks retrograde mitochondrial movement. Scale bar = 10 µm.

**Video 2.** Mitochondrial transport in a 4 dpf *actr10* mutant larva, related to *Figure 4*. Yellow arrowheads indicate mitochondria moving in the anterograde direction. Scale bar = 10 µm

## Dynein localization and motility does not rely on Actr10

Previous work on Actr10 function assayed mitotic spindle positioning during cell division in cultured fibroblasts and nuclear positioning in fungi to show that Actr10 depletion phenocopied dynein loss of function (*Lee et al., 2001*; *Yeh et al., 2012*; *Zhang et al., 2008*). In association with structural work implicating Actr10 in Arp1 filament capping (*Urnavicius et al., 2015*), these data could imply that loss of Actr10 impacts dynactin integrity and, subsequently, all dynein function. However, the aforementioned work could also be interpreted to mean that appropriate dynactin-nuclear membrane interaction requires Actr10 function. Supporting this possibility, Actr10 has been shown to regulate dynactin-membrane association (*Clark and Rose, 2006*). With this data in mind, we next wanted to determine if loss of Actr10 phenocopied dynein or dynactin loss of function in mature neurons, the cell type that

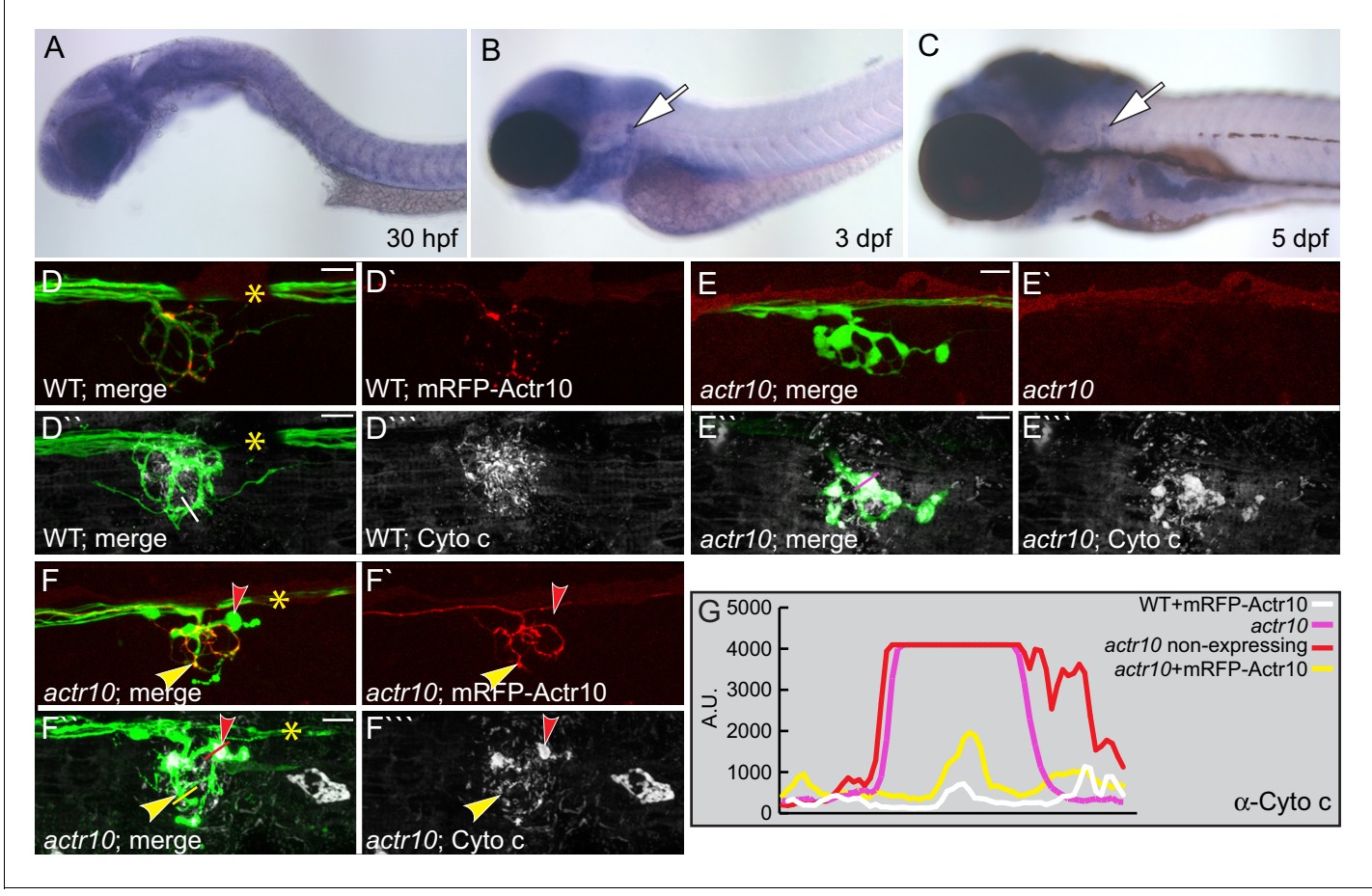

**Figure 5.** Actr10 functions cell autonomously in axons. (A–C) Expression of *actr10* mRNA was assayed using *in situ* hybridization with DIG-labeled riboprobes and alkaline phosphatase-mediated NBT/BCIP precipitation. *actr10* is ubiquitously expressed at all time points tested, with elevated expression in the nervous system at later stages (3 and 5 dpf). Arrows point to the pLL ganglion in B and C. (D) Neuronal mRFP-Actr10 expression does not alter axon morphology or mitochondrial localization in wildtype larvae at 4 dpf. Mitochondria were visualized by Cytochrome c immunolabeling in NM3 axon terminals. (E) Uninjected *actr10* mutants have high levels of Cytochrome c immunolabling in NM3 axon terminals at 4 dpf. (F) mRFP-Actr10 expression in *actr10* mutant neurons suppressed axon terminal swellings and mitochondrial accumulation. Yellow arrowhead points to the region of the axon terminal expressing mRFP-Actr10. Note that mRFP-Actr10 negative axon terminals (red arrowhead) display axonal swelling and high levels of Cytochrome c labeling. Lines in D'', E'' and F'' indicate regions used for line scan analysis. (G) Line scans show that mRFP-Actr10 expression rescues Cytochrome c levels in *actr10* mutant axons (compare pink/red and yellow lines). Pigment cells (*) obscure some nerves. Scale bars = 10 µm.

displayed specific defects in mitochondrial retrograde transport in our studies. First, we analyzed the localization of dynein and dynactin in *actr10* mutant axons at 4 dpf. Neither Dynein heavy chain (DHC) nor p150 were significantly mislocalized in *actr10* mutants, though we did observe a trend of increased DHC in *actr10* mutant axon terminals (*Figure 6A–F*). Next, to determine if dynein was able to move in the retrograde direction, we assayed dynein motility by tagging the core motor complex using expression of an mRFP tagged variant of dynein light intermediate chain (*5kbneurod: dync1li1V2-mRFP*; *Figure 6G,H*; *Videos 3* and *4*). Kymograph analysis revealed no change in the parameters of retrograde dynein movement, though we did note a 42% reduction in the distance and a 26% reduction in the velocity of *anterograde* dynein transport. Despite these defects in anterograde dynein transport, there was no change in the proportion of dynein-positive puncta moving in the anterograde or retrograde direction in *actr10* mutants at 4 dpf (*Figure 6I–M*). Additionally, we analyzed reversal frequency to determine if there was a defect in persistence of dynein movement in *actr10* mutants but found no difference in the average number of puncta that reversed direction during our imaging sessions (wildtype: 1.62/100 μm*min ±0.37; *actr10* mutants: 1.74/100 μm*min ±0.44; ANOVA; *p*=0.844). These experiments demonstrated that loss of Actr10 does not impede dynein-dynactin complex localization or dynein retrograde movement in axons.

We then analyzed the stability of dynein-dynactin interaction using co-immunoprecipitation from whole larval extracts. Immunoprecipitation of endogenous proteins was not possible due to lack of zebrafish-specific antibodies, so we turned to overexpression of tagged dynein components using zygotic injection of *in vitro* synthesized mRNA. First, we confirmed that a GFP tagged version of dynein intermediate chain 2b (i2b), a core dynein protein, could integrate into the motor complex. mRNA encoding i2b-GFP was injected into zygotes and anti-GFP antibodies were used to immunoprecipitate the complex. Extracts were subjected to western blot analysis and probed with anti-DHC antibodies. i2b-GFP can integrate into the core dynein complex (*Figure 6N*). We then used this approach to determine if loss of Actr10 impacts dynein-dynactin interaction. i2b-GFP was expressed in *actr10* mutants and wild-type siblings and larval extracts were subjected to GFP-based immunoprecipitation at 4 dpf. Western blot of larval extracts revealed no change in p150 interaction with the core dynein motor (*Figure 6O*). Together, these experiments confirmed that loss of Actr10 does not impact dynein-dynactin interaction.

To further explore the effect of loss of Actr10 on retrograde cargo transport, we analyzed the movement of additional dynein cargos in axons. First, we visualized peroxisome transport in lateral line axons (*Videos 5* and *6*). Kymograph analysis of peroxisome transport at 4 dpf demonstrated small changes in anterograde transport velocity and distance of retrograde transport bouts (*Figure 7A–G*). A slight, but not significant decrease in the proportion of peroxisomes moving in the retrograde was apparent as well (*Figure 7G*; ANOVA; *p*=0.07). Upon further investigation, we found that previous studies revealed a surprising relationship between mitochondria and peroxisomes, including a large number of shared membrane proteins, such as Drp1 (reviewed in *Schrader, 2006*). Our work implicates Drp1 in Actr10-dependent mitochondrial localization (see Figure 12), making the impact of Actr10 loss of function on peroxisome transport complex. Therefore, we analyzed the transport of a third cargo, Lamp1-labeled vesicles, which are composed of late endosomes and lysosomes. Analysis of Lamp1-positive vesicle movement at 4 dpf demonstrated that this cargo is transported normally in *actr10* mutant axons (*Figure 7H–L*). These live imaging experiments, in conjunction with the immunolabeling of known retrograde cargos (*Figure 2*) and our TEM studies (*Figure 3*), argue that loss of Actr10 does not hinder the transport of all dynein-dependent, retrograde cargos in axons.

Since we observed small changes in retrograde peroxisome transport, we wanted to further investigate the relationship between loss of Actr10 and loss of dynein/dynactin. To do this, we analyzed the ability of DHC and p150 null mutants to phenocopy *actr10* mutants. First, we compared *actr10* mutants to a previously isolated zebrafish DHC mutant, *dync1h1^{mw20}* (hereafter referred to as *dync1h1*; *Insinna et al., 2010*). Unlike *actr10* mutants (see *Figure 1*), *dync1h1* mutants displayed rapid degeneration of pLL axons by 4 dpf, with pLL nerves extending less than half-way to the tail at this time-point (*Figure 8A,B*). Mitochondria do accumulate in distal axons of *dync1h1* mutants, as expected with global disruption of retrograde transport (*Figure 8F,G*). The disparity between *actr10* and *dync1h1* mutant axonal phenotypes argues that loss of Actr10 does not impact all dynein function in axons.

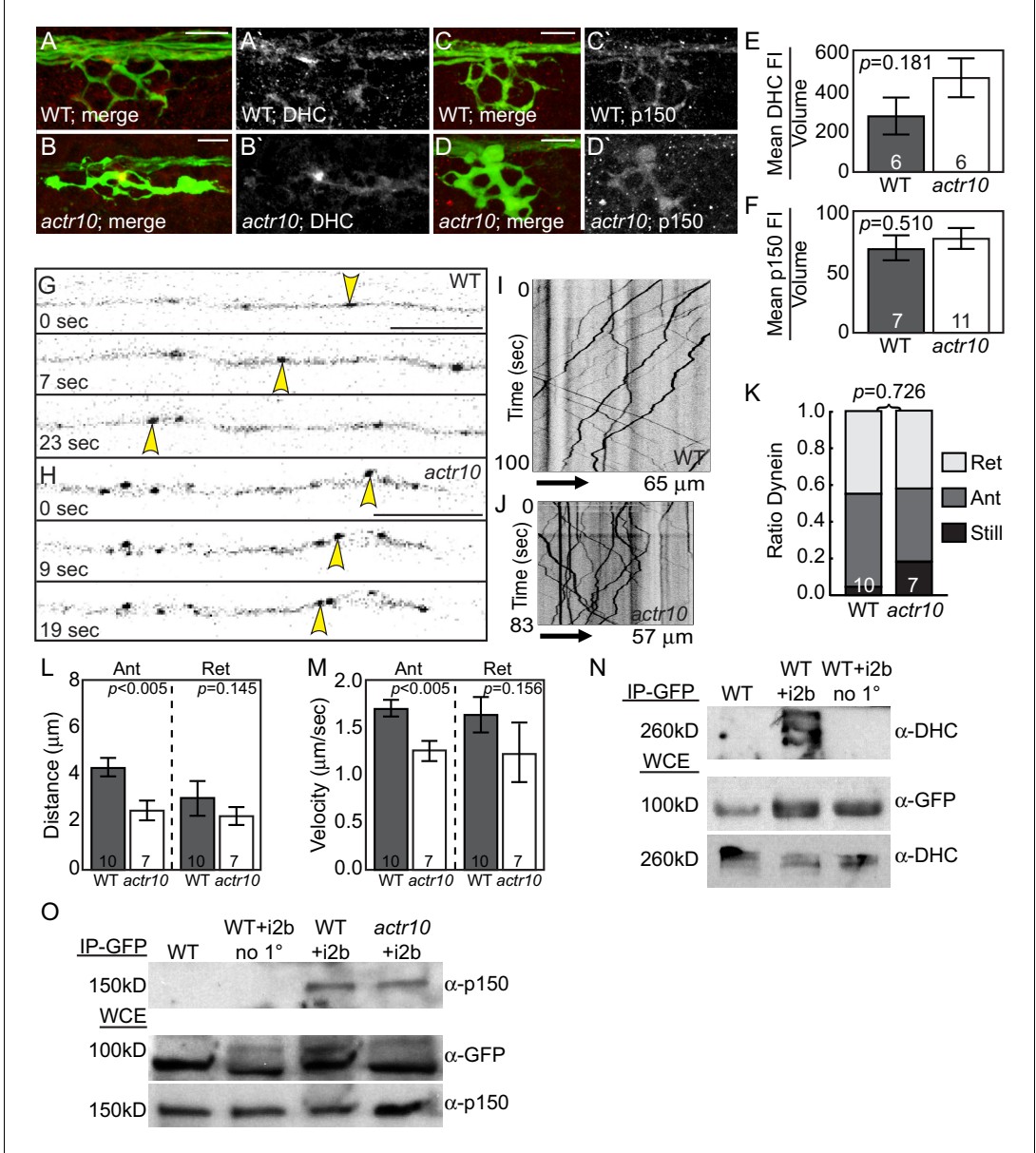

**Figure 6.** Dynein-dynactin localization and retrograde movement are intact in *actr10* mutants. (A,B) Dynein heavy chain (DHC) immunolabeling demonstrates normal DHC localization in NM3 axon terminals of *actr10* mutants at 4 dpf. (C,D) p150 is normally localized in NM3 axon terminals of *actr10* mutants at 4 dpf. (E,F) Analyses of mean fluorescence intensity in axon terminals (background subtracted) showed comparable fluorescence intensity between mutant and wildtype siblings (ANOVA; mean ± SEM shown). (G,H) Stills from dynein time-lapse imaging sessions (*Videos 3* and *4*) in wildtype and *actr10* mutant pLL axons at 4 dpf. Arrowheads indicate retrograde dynein movement. (I,J) Kymograph analyses of dynein transport in wildtype (I) and *actr10* mutants (J). (K–M) Retrograde dynein transport parameters, including the proportion of dynein-labeled vesicles (K), distance moved by vesicles (L) and velocity of movement (M) are unaffected in *actr10* mutants (ANOVA; mean ± SEM shown). A reduction of anterograde dynein-positive puncta distance and velocity was noted (ANOVA; mean ± SEM shown). (N) Dynein intermediated chain 2b fused to GFP (i2b-GFP) interacts with the core dynein complex. Immunoprecipitation of i2b-GFP from whole embryo extracts followed by DHC western (top). Whole embryo lysate controls for i2b-GFP (middle; top band not present in wildtype) and DHC (bottom). Bead only (no 1°) immunoprecipitation controls shown. (O) The core dynein complex labeled by i2b-GFP can immunoprecipitate p150 from whole larval lysates derived from wildtype and *actr10* mutant larvae at 4 dpf (top). Whole larval lysate control for i2b-GFP (middle) and p150 (bottom). Bead only (no 1°) and un-injected wildtype immunoprecipitation controls shown. Scale bars = 10 µm. Number of larvae assayed is indicated on graphs.

**Video 3.** Dynein motility in a 4 dpf wildtype larvae, related to *Figure 6*. Red arrowheads indicate dynein positive cargo moving in the retrograde direction. Scale bar = 10 μm

**Video 4.** Dynein motility in a 4 dpf *actr10* mutant larva, related to *Figure 6*. Yellow arrowhead indicates dynein positive puncta being passively carried in the anterograde direction. Red arrowheads mark dynein positive cargo moving in the retrograde direction. Scale bar = 10 μm

We then compared the *actr10* mutant phenotype to loss of dynein-dynactin interaction using zebrafish *p150* mutants. There are two orthologs of *p150* in zebrafish, *p150a* and *p150b*. These paralogues are highly similar at the amino acid level but have slightly different expression patterns: while both are ubiquitously expressed, *p150b* is enriched in developing (1 and 2 dpf) and mature (4 dpf) neurons while *p150a* is enriched only in mature neurons 4 dpf (*Figure 9A–F*). To determine whether loss of p150 phenocopies the *actr10* mutant, we assayed single and double *p150* mutants using a *p150a* mutant described previously (*Del Bene et al., 2008*; *Wehman et al., 2005*) and a novel *p150b* mutant we engineered (*p150b^{nl16}*). Loss of *p150a* alone has no discernable effect on axons or mitochondrial localization (*Figure 8C,H*). We used CRISPR-Cas9 technology (*Hwang et al., 2013*) to create a deletion in exon 3 of *p150b*, which resulted in a frame-shift and premature stop site. *p150b* single mutants are indistinguishable from wild-type siblings at 4 dpf, with normal axon terminals and no abnormalities in mitochondrial localization (*Figure 8D,I*). This strain does not have the small eye phenotype observed in *dync1h1* and *p150a* mutants (*Figure 8A–D*; *Del Bene et al., 2008*; *Insinna et al., 2010*; *Wehman et al., 2005*). Unlike single *p150* mutants, *p150a/b* mutants are phenotypically similar to *dync1h1* mutants, with truncated pLL axons and mitochondrial accumulation at axon ends (*Figure 8E,J*). This confirms the reliance of mitochondria on the dynein-dynactin complex for retrograde movement and demonstrates that *actr10* mutants do not phenocopy the axon truncation phenotype of dynein or dynactin loss of function mutants.

We then analyzed the effect of p150 loss of function on mitochondrial transport in pLL axons. Mitochondria were tagged with TagRFP using zygotic injection of the *5kbneurod:mito-TagRFP* plasmid and kymograph analysis was performed on videos acquired through live imaging (*Figure 8K–N*). Unlike loss of Actr10, *p150a/b* double mutants had an almost complete cessation of mitochondrial movement in the anterograde and retrograde directions (*Figure 8N,O*). In addition, we noted a dramatic loss of mitochondria from the central portion of axons in *p150a/b* double mutants, likely because of the strong defect in anterograde mitochondrial transport (*Figure 8P*). Analyses of transport parameters revealed no change in the distance or velocity of the residual mitochondrial movement in either direction (*Figure 8Q,R*). The dramatic loss of all mitochondrial movement in *p150a/b* mutants, specifically reductions in both the anterograde and retrograde pools, is not observed in *actr10* mutants, which have only a reduction in the retrograde mitochondrial pool (see *Figure 4*). Furthermore, two cargos known to localize normally in *actr10* mutants, pJNK and Lamp1 (see *Figure 2*), accumulate in in *p150a/b* mutant axon terminals (*Figure 10A–D*), indicating that the retrograde transport of these cargos is perturbed by loss of dynein-dynactin interaction. Therefore, our

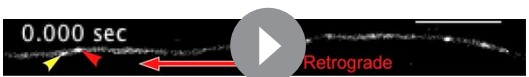

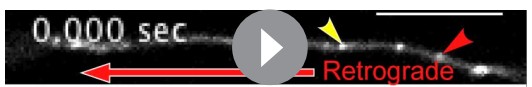

**Video 5.** Peroxisome motility in a 4 dpf wildtype larvae, related to *Figure 7*. Yellow arrowhead points to a peroxisome moving in the anterograde direction. Red arrowhead indicates a peroxisome moving in the retrograde direction. Scale bar = 10 μm

**Video 6.** Peroxisome motility in a 4 dpf *actr10* mutant larva, related to *Figure 7*. Yellow arrowheads indicate peroxisomes moving in the anterograde direction. Red arrowhead marks peroxisome moving in the retrograde direction. Scale bar = 10 μm

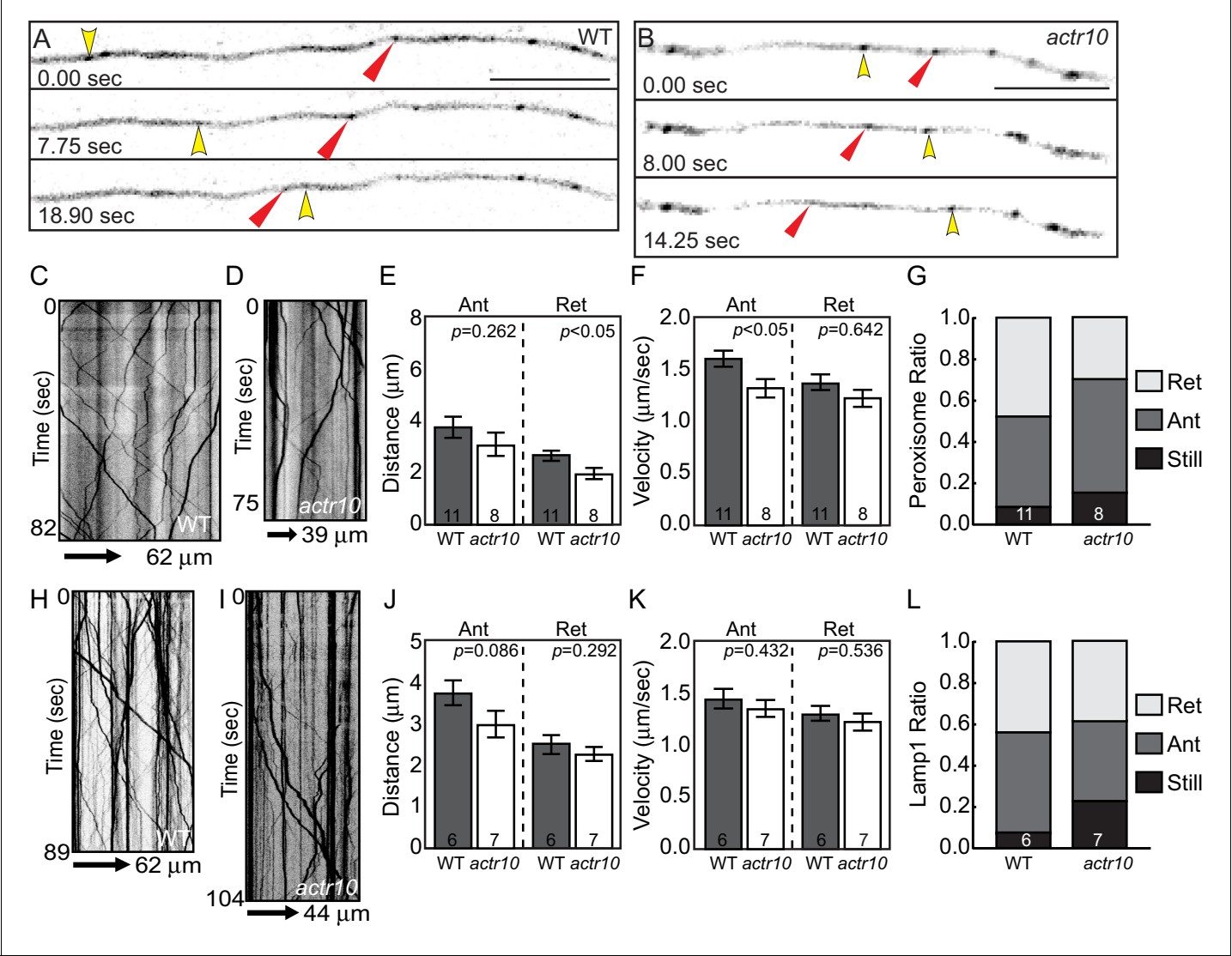

**Figure 7.** Peroxisome and Lamp1 vesicle transport in *actr10* mutants at 4 dpf. (A,B) Stills from peroxisome imaging sessions at 4 dpf from *Videos 5* and *6*. Yellow and red arrowheads point to anterograde and retrograde peroxisome movement respectively. (C,D) Kymographs analyses of peroxisome transport in wildtype (C) and *actr10* mutants (D). (E–G) Peroxisome transport parameters were similar between *actr10* mutants and wildtype siblings, though a slight reduction in retrograde distance and anterograde velocity were noted (ANOVA; mean ± SEM shown). The proportion of peroxisomes moving in the retrograde direction was reduced as well, though not significantly (ANOVA; mean ± SEM shown; *p*=0.07). (H,I) Kymographs of Lamp1-labeled vesicle movement at 4dpf. (J–L) Lamp1 vesicles were transported similarly between wildtype and *actr10* mutants. An increase in the proportion of these vesicles in mutants that were stationary during imaging was noted (*p*<0.01; ANOVA; mean ± SEM shown). Number of larvae assayed is indicated on graphs. Scale bars = 10 µm.

phenotypic, *in vivo* transport, immunolabeling and biochemical data fail to support the argument that loss of Actr10 impacts all dynein-dynactin function in axons and, instead, supports a specific requirement for Actr10 in mitochondrial retrograde transport in this neuronal compartment.

## Mitochondria fail to attach to the dynein-dynactin complex in actr10 mutants

The specific effect of Actr10 depletion on retrograde mitochondrial transport frequency led us to ask if Actr10 is necessary to attach this organelle to the dynein motor complex. We addressed this question using mitochondrial fractionation from whole larvae (*Figure 11A*; *Prudent et al., 2013*).

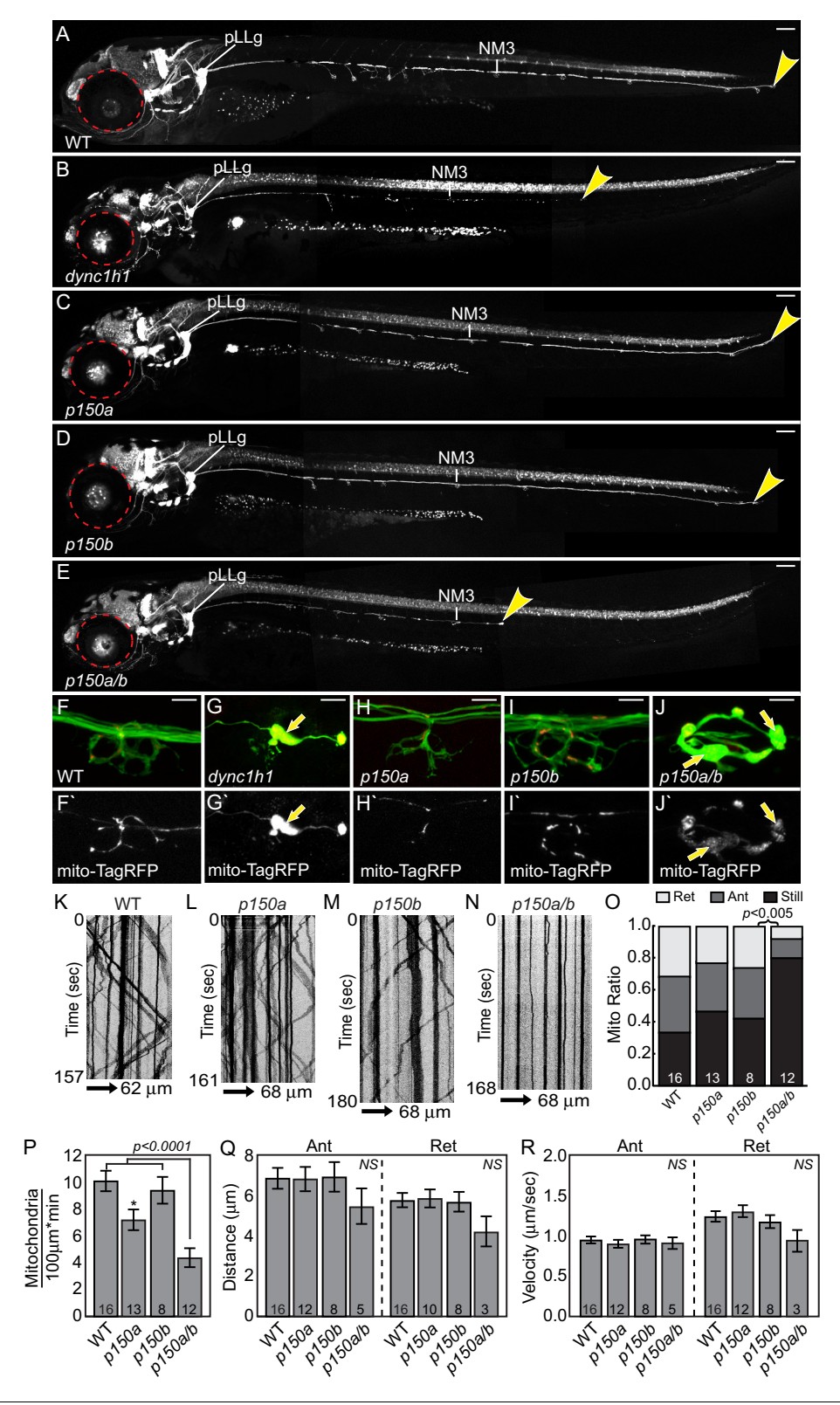

**Figure 8.** *dync1h1* and *p150a/b* mutants fail to phenocopy *actr10* mutants. (**A**) Wildtype larva at 4 dpf with full pLL axon extension into the tail. End of pLL nerve indicated by arrowhead. (**B**) Loss of DHC in *dync1h1* mutants results in pLL axon degeneration, leading to a truncated pLL nerve at 4 dpf. Small eyes are also apparent (red dotted line), as has been reported previously. (**C**) *p150a* mutants have small eyes (red outline) but pLL nerves are identical to wildtype siblings. (**D**) *p150b* mutants at 4 dpf with normal pLL nerves and no change in eye diameter. (**E**) *p150a/b* double mutants have

*Figure 8 continued on next page*

*Figure 8 continued*

truncated, thin pLL nerves with swellings along the length of axons and in terminals and are indistinguishable from *dync1h1* mutants. *p150a/b* mutants also show small eyes similar to *dync1h1* and *p150a* mutants. (F,G) *dync1h1* mutants display axon terminal swellings and mitochondrial accumulation compared to wildtype controls at 4 dpf. Arrows point to mitochondria rich, axonal swellings. Mitochondria are labeled by expression of *mito-TagRFP* (red on top F-J, white below F'-J'). (H,I) *p150a* and *p150b* single mutants have axon terminal morphology and mitochondrial localization in axon terminals similar to wildtype siblings. (J) Loss of both p150 paralogues (*p150a/b*; J) results in swollen axon terminals with increased mitochondrial density similar to *dync1h1* mutants (G). (K–N) Kymograph analyses of mitochondrial transport in *p150* mutants. (O) The proportion of mitochondria moving in both the anterograde and retrograde direction in pLL axons is significantly decreased in *p150a/b* double mutants (ANOVA with Tukey HSD post-hoc contrasts). (P) The number of mitochondria in pLL axons is decreased in *p150a* mutants compared to wildtype siblings (ANOVA with Tukey HSD post-hoc contrasts; *$p<0.05$). The number of mitochondria in pLL axons of *p150a/b* double mutants is dramatically reduced compared to wildtype siblings (ANOVA with Tukey HSD post-hoc contrasts; mean±SEM shown). (Q,R) Distance and velocity of limited residual mitochondrial movement is not significantly altered in *p150a/b* mutants compared to wildtype siblings (ANOVA with Tukey HSD post-hoc contrasts; mean ± SEM shown). Scale bars in A-E = 100 µm. Scale bars in F-J = 10 µm. Number of larvae assayed is indicated on graphs.

Analysis of fractions revealed equal levels of p150 in the input and heavy fractions between wildtype and *actr10* mutants. Strikingly, in *actr10* mutants, p150 was largely lost from the mitochondrial fraction with a concomitant increase of p150 to the light fraction, which contains all cellular components not pelleted under low centrifugation speeds (*Figure 11B,C*). This result demonstrates the necessity of Actr10 for mitochondrial attachment to dynactin.

To better define the mechanism of dynactin-Actr10-mitochondrial interaction, we identified the domain in Actr10 necessary for association with the dynactin complex using immunoprecipitation of deletion constructs (overlapping deletion of 40 amino acid regions) from HEK293T cells. Whereas Actr10△7–△10 deletions showed reduced binding to dynactin, Actr10△7 most consistently failed to immunoprecipitate dynactin in all experiments (*Figure 11E,F*; n = 4). As this result was somewhat variable in HEK cells, we confirmed the necessity of this domain for Actr10-dynactin interaction *in vivo* using immunoprecipitation of Actr10 and Actr10△7 from embryo lysates. This revealed a loss of interaction between Actr10△7 and dynactin, demonstrating the reliance of Actr10 on this region for interaction with dynactin (*Figure 11G*; n = 3). Finally, if Actr10 is a part of the scaffold necessary for mitochondria-dynactin interaction, a separate mitochondrial binding domain likely exists in Actr10. If this is the case, Actr10 lacking the dynactin binding domain should maintain its interaction with mitochondria. To address this, we assayed the ability of Actr10△7 to bind mitochondria using mitochondrial fractionation. mRFP-Actr10 and mRFP-Actr10△7, expressed using zygotic microinjection of respective mRNAs, were both present in the mitochondrial fractionation (*Figure 11D,H*; n = 2), arguing that separate protein domains exist in Actr10 which are necessary for interaction with dynactin and mitochondria. This data supports a role for Actr10 in binding mitochondria to dynactin for retrograde mitochondrial transport.

## Drp1 functions with Actr10 in mitochondrial retrograde transport

It is unlikely that Actr10 binds directly to mitochondria as it does not have predicted transmembrane or other membrane-associated domains. Rather, similar to Kinesin-1-mitochondrial attachment, there are likely partner proteins that facilitate this interaction (*Glater et al., 2006*; *Guo et al., 2005*). Literature searches revealed a particularly strong association between Drp1 (Dynamin related protein 1), a GTPase associated with mitochondrial fission, and mitochondrial localization. Specifically, a lysine to alanine mutation at amino acid 38 in Drp1 (mimicking a constitutively GDP-bound form) causes clustering of mitochondria in the perinuclear region of fibroblasts in a microtubule-dependent manner (*Smirnova et al., 2001*; *Varadi et al., 2004*). Using immunoprecipitation, we demonstrated that Drp1 interacts with Actr10, with strongly enhanced interaction between Actr10 and the Drp1$^{K38A}$ variant (*Figure 12A*; n = 3 replicates in independent experiments; (*Smirnova et al., 2001*, *1998*).

Next, we asked whether Drp1 works with Actr10 to regulate mitochondrial retrograde transport *in vivo*. To address this, we examined the genetic interaction between Drp1 and Actr10 in mitochondrial localization in pLL neurons. We reasoned that if Drp1 and Actr10 function in the same pathway, Drp1 manipulation would impact the location of mitochondria in wildtype but not *actr10* mutant axons. We expressed Drp1-mRFP, Drp1$^{K38A}$-mRFP, and cytoplasmic mRFP (control) in pLL neurons using injection of a plasmid encoding the respective open reading frame under an inducible

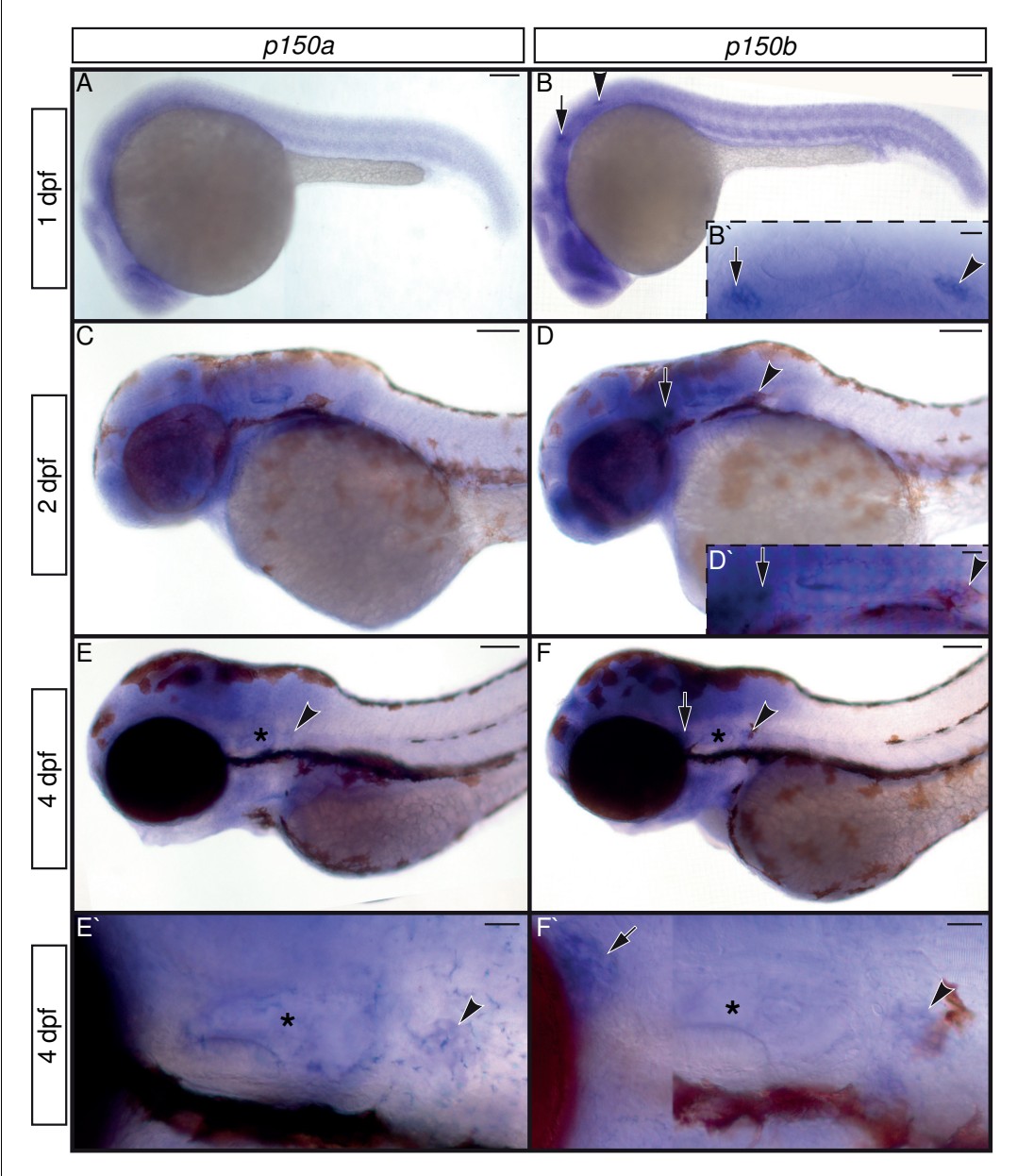

**Figure 9.** *p150a* and *p150b* expression in zebrafish embryos and larvae. (A,C,E,E') *p150a in situ* hybridization shows ubiquitous expression at 1 and 2 dpf, with a slight enrichment in neurons of the pLL ganglion at 4 dpf. (B,D,F,F') *p150b* is enriched in developing neurons with low level expression ubiquitously at all stages assayed. pLL and anterior lateral line ganglia are indicated by an arrowhead or arrow respectively in all. *=ear. Higher magnification views of the ear, pLL ganglia, and anterior lateral line ganglia shown in insets in B', D', E', and F'. Scale bars in A-F = 100 μm. Scale bars in B' and D'-F'=20 μm.

promotor. Mitochondria were labeled by *5kbneurod:mito-EGFP* co-injection. In wild-type larvae, expression of Drp1$^{K38A}$-mRFP caused mitochondrial clustering in the perinuclear region, phenotypically similar to the results observed in cultured fibroblasts (*Figure 12B,C*; *Smirnova et al., 2001*; *Varadi et al., 2004*). In contrast, Drp1$^{K38A}$-mRFP expression did not alter mitochondrial localization in *actr10* mutant pLL neurons (*Figure 12D,E*). If Drp1$^{K38A}$ mediates retrograde mitochondrial movement, we predicted its expression would induce movement of mitochondria from the proximal axon into the cell body, leading to loss of mitochondrial occupation of the axon. Indeed, expression of Drp1$^{K38A}$ caused a substantial decrease in the number of mitochondria per micron in wildtype but

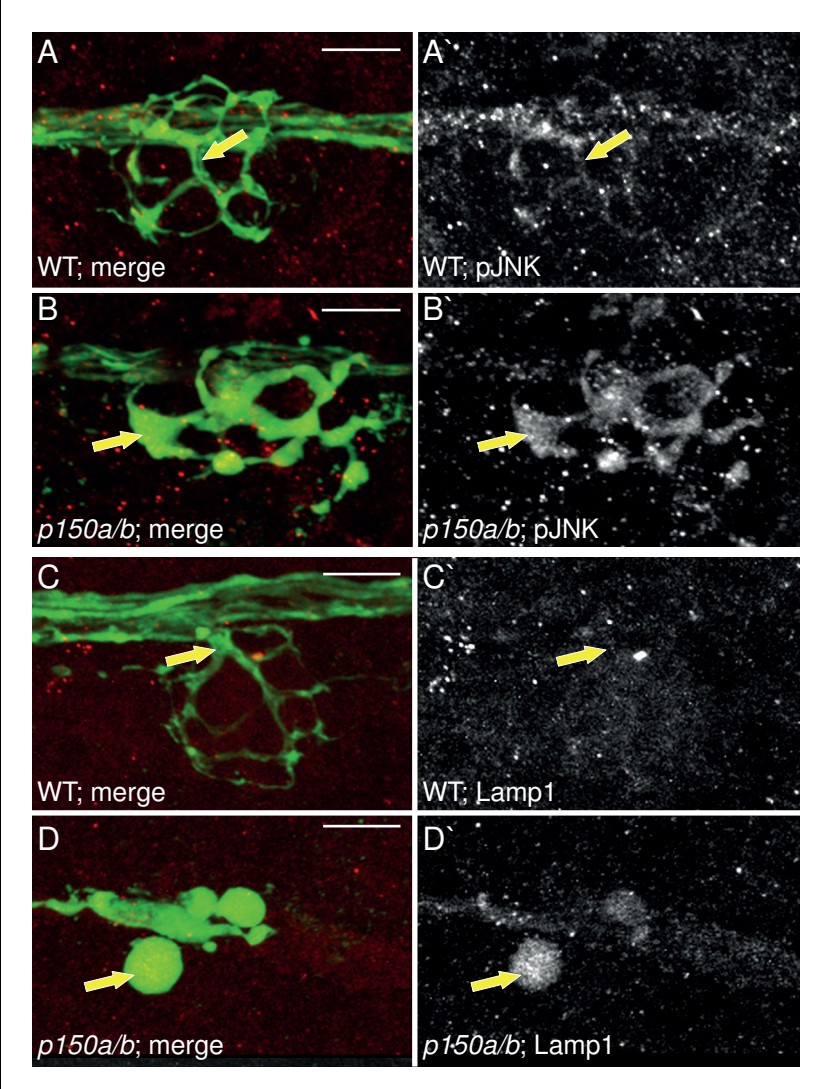

**Figure 10.** pJNK and Lamp1 accumulate in *p150a/b* pLL axon terminals. (**A,B**) pJNK accumulates in axon terminals at 4 dpf in *p150a/b* double mutants. NM3 shown. (**C,D**) Lamp1 accumulates in NM4 axon terminals at 5 dpf in *p150a/b* double mutants. Arrows point to axon terminals in wildtype and *p150a/b* double mutants in all. Axons are labeled by the *neurod:egfp* BAC transgene (green). Lamp1 and pJNK are show in red in A-D and white in A'-D'. Scale bars = 10 µm.

not *actr10* mutant axons (**Figure 12F–J**). Notably, *actr10* mutants already have reduced mitochondrial occupation of pLL axons which could impact the fraction of mitochondria capable of motility in this assay. More definitive analyses of Drp1's role in Actr10-dependent mitochondrial motility will be the subject of continued investigation. Together, the biochemical and genetic interaction between Actr10 and Drp1$^{K38A}$ support a model in which Drp1, in its GDP-bound state, modulates dynein-based mitochondrial localization via an Actr10-dependent mechanism.

## Discussion

Using genetics, immunolabeling, biochemistry, and *in vivo* imaging of cargo movement, we have identified Actr10 as a mediator of retrograde mitochondrial transport. In the absence of this dynactin pointed end protein, mitochondrial retrograde transport frequency is selectively disrupted, leading to accumulation of this organelle in axon terminals. Importantly, the anterograde transport of

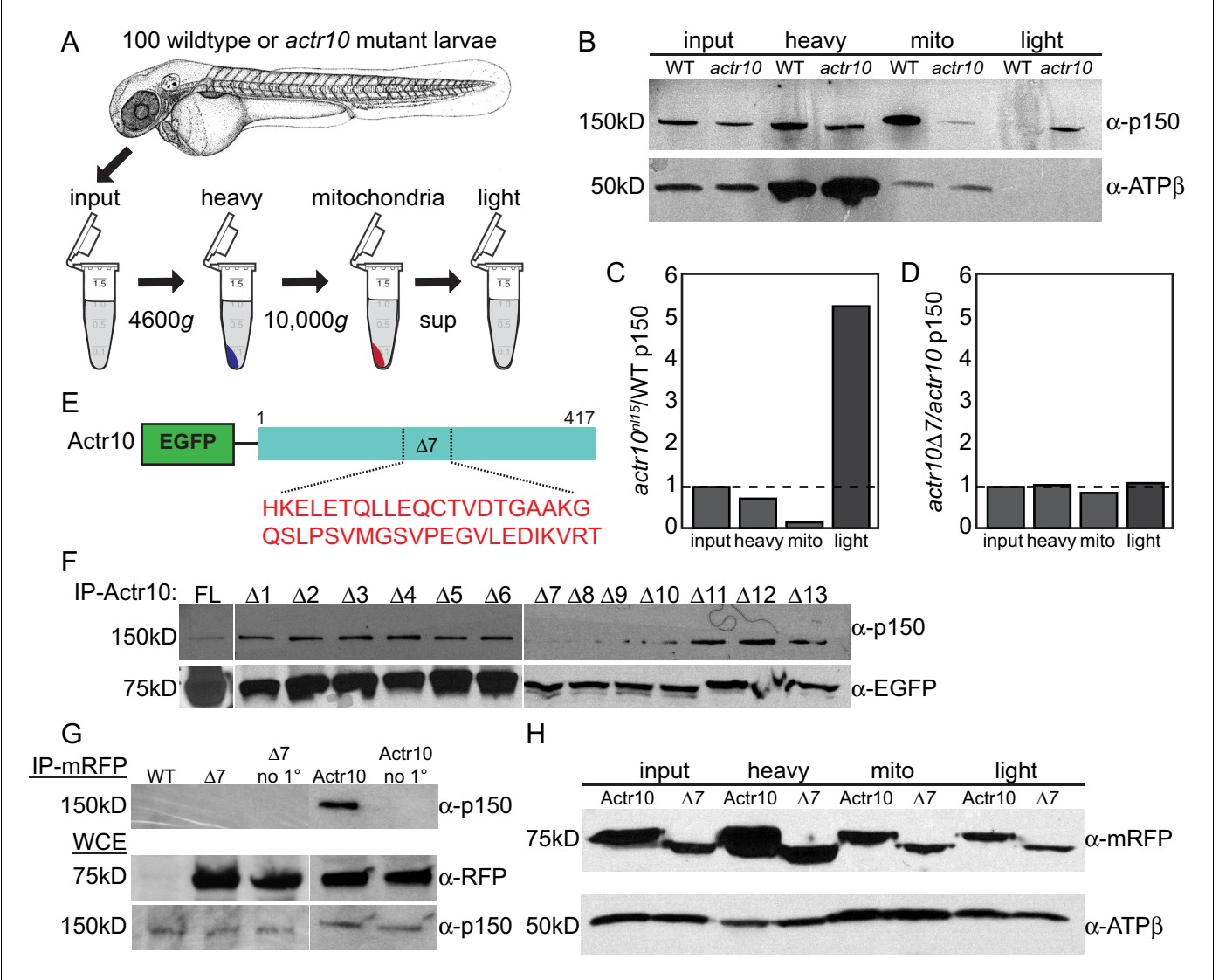

**Figure 11.** Actr10 is essential for mitochondria-dynactin interaction. (**A**) Schematic of whole embryo mitochondrial fractionation. (**B**) While the level of p150 in the input and heavy fractions are comparable between wildtype and *actr10* mutants, p150 is shifted from the mitochondrial to light fraction in *actr10* mutants. ATPβ western blot from the same extracts (bottom) serves as a mitochondrial loading control. (**C**) Quantification of p150 intensity (normalized to mitochondrial loading) in *actr10* mutants relative to wildtype. (**D**) Quantification of p150 intensity (normalized to mitochondrial loading) in mRFP-Actr10△7 expressing embryos relative to those expressing mRFP-Actr10. (**E**) Schematic and sequence of the △7 region in Actr10. (**F**) Immunoprecipitation of EGFP-Actr10 deletion constructs from HEK293T cells identified the △7 region as critical for Actr10's interaction with the dynactin complex (n = 4 replicates). (**G**) In zebrafish embryos, mRFP-Actr10△7, expressed by zygotic injection of *in vitro* synthesized mRNA, was unable to interact with dynactin (top). mRFP-Actr10 and mRFP-Actr10△7 (middle) and p150 (bottom) are all present at similar levels. Bead only (no 1°) immunoprecipitation controls shown. (**H**) Actr10△7 is able to interact with mitochondria in zebrafish embryos as assayed by mitochondrial fractionation. ATPβ mitochondrial loading control (bottom).

mitochondria, the localization and transport of other cargos assayed, the localization of dynein-dynactin components as well as dynein retrograde movement are all largely intact in the absence of Actr10, arguing for a role for Actr10 in mitochondrial retrograde transport in axons. Additionally, Actr10 maintains its ability to interact with mitochondria without being incorporated into the dynactin complex, suggesting a specific role for this protein in linking this organelle to the retrograde motor. Finally, we demonstrated that Actr10, perhaps through interaction with Drp1, links mitochondria to the dynactin complex. Our study brings us a step closer to understanding how unidirectional mitochondrial transport is mediated in axons to maintain a functional neural circuit.

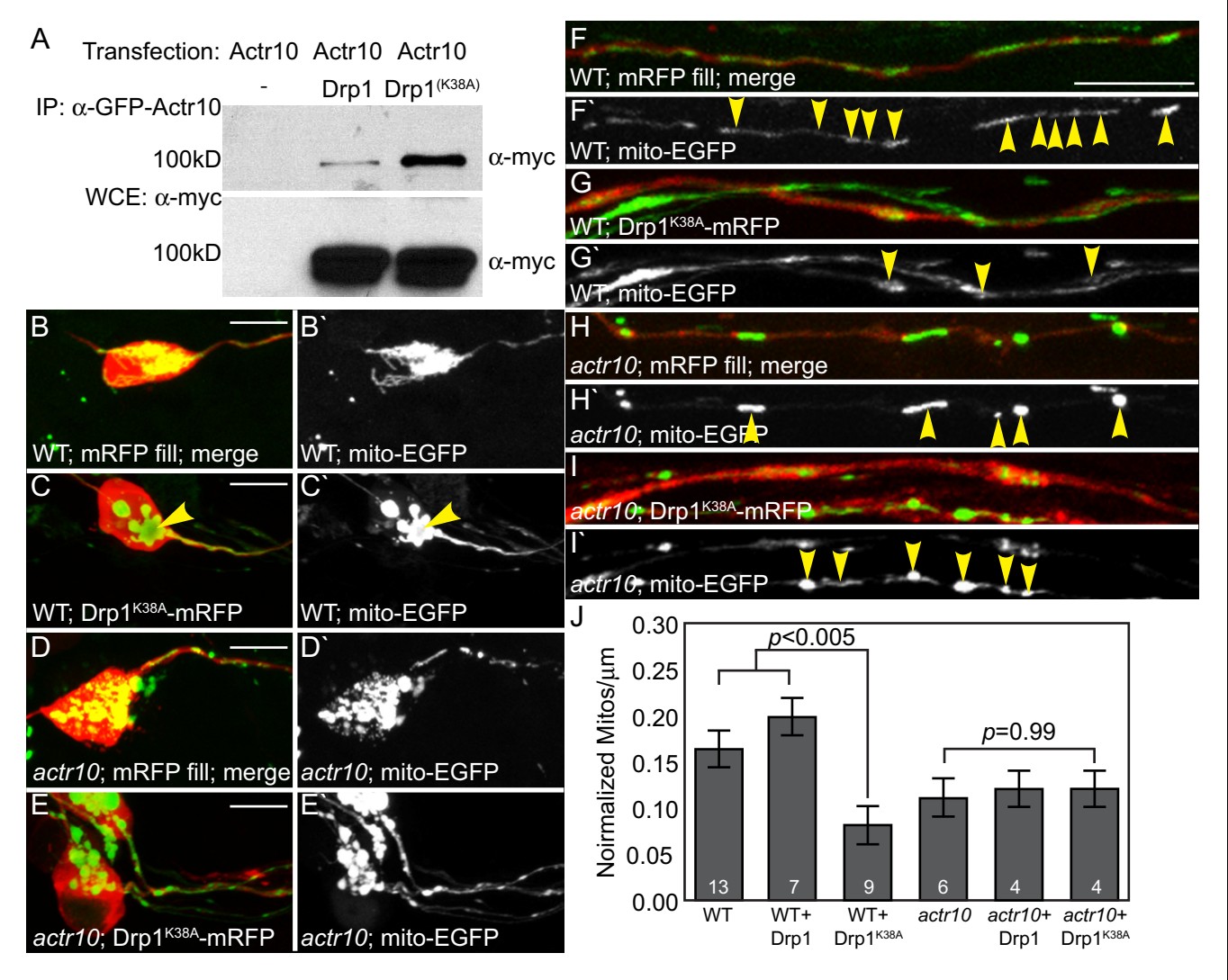

**Figure 12.** Drp1 functions with Actr10 in mitochondrial retrograde transport. (**A**) EGFP-Actr10 interacts with myc-Drp1 in HEK293T cells. The K38A mutation in Drp1 strengthens this interaction (n = 3 replicates). (**B,C**) Expression of Drp1^K38A-mRFP causes mitochondrial clustering in wildtype pLL neuron cell bodies at 4 dpf compared to an mRFP only control (**B**). (**D,E**) Though mitochondrial morphology is abnormal in *actr10* mutant pLL neuron cell bodies (**D**), expression of Drp1^K38A-mRFP does not cause clustering of this organelle (**E**). (**F–I**) While expression of Drp1^K38A-mRFP in wildtype larvae causes loss of mitochondria from the proximal axon compared to mRFP only controls (**F,G**), mitochondrial number is unaffected by Drp1^K38A-mRFP expression in *actr10* mutants (**H,I**). Arrowheads point to mitochondria in mRFP-expressing axons. Mitochondria are labeled by expression of a mitochondrial targeting sequence tagged with EGFP (*5kbneurod:mito-EGFP*; green in B-I, white in B'-I'). (**J**) Quantification of mitochondrial number normalized to mitochondrial size in the proximal axon (ANOVA with post-hoc contrasts; mean ± SEM shown). Scale bars = 10 µm. Number of larvae assayed is indicated on graphs.

## Actr10s role in mitochondrial retrograde transport

We provide multiple pieces of evidence supporting a function for Actr10 in mitochondrial transport. First, mitochondria are the only cargo analyzed whose localization or retrograde transport is significantly perturbed with loss of Actr10. If dynein-mediated cargo movement was generally disrupted, other cargos, including lysosomes and dynein itself, would fail to move in the retrograde direction as well. Second, the axonal phenotype of dynactin and dynein loss-of-function mutants is vastly different from *actr10* mutants: *dync1h1* and *p150a/b* mutants display pLL nerve degeneration at 4 dpf while *actr10* mutants do not. Third, loss of dynactin-dynein interaction in *p150a/b* mutants impedes all mitochondrial transport, a phenotype that is not observed in *actr10* mutants. Fourth, neither the

localization of dynein and dynactin nor the retrograde movement of dynein are perturbed with loss of Actr10. Lastly, we identified a domain in Actr10 (△7) that is essential for interaction with the dynactin complex but inconsequential for interaction with mitochondria. This piece of data is critical as it demonstrates that Actr10 can interact with this organelle independently of dynactin, arguing for a direct role for this protein in mitochondria-dynactin interaction. Together, our data argue that Actr10 is necessary for dynactin-dependent, retrograde transport of mitochondria in axons.

Actr10 could directly link mitochondria to dynactin or potentially facilitate this interaction through other dynactin pointed end proteins, such as p62. p62 is of particular interest in this regard as it is predicted to rely on Actr10 for binding to the dynactin complex (*Urnavicius et al., 2015*). Based on our current data, we cannot rule out a role for this protein in mitochondrial transport. Furthermore, Actr10 interacting proteins, including p62, could have additional impacts on mitochondrial transport parameters, including transport distance. As noted above, *actr10* mutants have slight deficits in the distance moved by mitochondria in both the anterograde and retrograde directions. While we cannot definitively ascertain the underlying mechanism of this disruption, at least three possibilities exist. First, bidirectional transport distance could be affected by the abnormal localization of mitochondria. Mitochondrial localization directly affects local levels of ATP, which in turn could compromise the processive activity of ATP-dependent motors necessary for long distance transport. Another possibility is local deficits in mitochondrial activity: if mitochondrial health is compromised, this could account for the bidirectional transport deficits of mitochondria. Finally, it is possible that Actr10 or its interacting proteins, such as p62, regulate motor engagement. It is largely thought that cargos typically have anterograde and retrograde motors attached simultaneously and direction of transport is attributed to regulation of motor engagement. More frequent oscillations in motor engagement due to loss of Actr10 or Actr10 interactors could account for the shortened distances moved by mitochondria in both directions. At this point, we do not have a full explanation for these transport parameter alterations but they are a topic of continued interest.

## Multiple roles for Actr10 in the cell

Our data does not preclude a function for Actr10 in the retrograde transport of additional cargos in axons or other roles for Actr10, which could vary based on a number of factors including cell type and developmental stage. Of particular interest is a potential role for Actr10 in peroxisome transport through interaction with Drp1. While we did not find a statistically significant difference in the proportion of peroxisomes moving in the retrograde direction, there was a trend towards a decrease in this population. As Drp1 is known to localize to peroxisome membranes as well (*Schrader, 2006*), it will be interesting to further explore the interplay between these proteins and the transport of these highly related organelles. In addition to cargo transport, one predicted function for Actr10, based on structural and biochemical data, is in capping the Arp1 filament (*Urnavicius et al., 2015*). In support of this role, knockdown of Actr10 using siRNA in Cos7 cells resulted in dissociation of the dynactin complex pointed end (*Yeh et al., 2012*). Additionally, studies in the fungi *neurospora* and *aspergillus* demonstrated that loss of Actr10 phenocopies the abnormal nuclear positioning observed with loss of dynein (*Lee et al., 2001*; *Zhang et al., 2008*). Similarly, zebrafish *actr10* mutants display the small eye phenotype observed in *dync1h1* and *p150a* mutants (see *Figure 8* and *Del Bene et al., 2008*; *Insinna et al., 2010*). Finally, similar to loss of p150, knockdown of Actr10 resulted in multipolar spindle formation in Cos7 cells (*Yeh et al., 2012*). Together, these data could imply that Actr10 is in fact necessary for all dynein-dynactin activity through stabilizing the Arp1 filament; however, they could also be interpreted to mean that Actr10 is necessary for regulation of Arp1-membrane interaction. Indeed, Actr10 has been shown to regulate Arp1-membrane interaction (*Clark and Rose, 2006*) and the zebrafish eye phenotype in the *p150a* mutant is due to a defect in interkinetic nuclear migration (*Del Bene et al., 2008*). This type of function for Actr10 could also explain the nuclear positioning defect in the *actr10* loss of function studies in fungi (*Lee et al., 2001*; *Zhang et al., 2008*). Therefore, rather than purely serving a stabilizing role for the dynactin complex, Actr10 may modulate both Arp1 interaction with the nuclear membrane and mitochondria-dynactin binding in axons. How these functions are disparately regulated in different cell types, in different cellular compartments and/or at varying developmental stages is still unclear.

Given the previous data implicating Actr10 in Arp1 capping and subsequent dynactin stability and dynein function (*Urnavicius et al., 2015*; *Yeh et al., 2012*; *Zhang et al., 2008*), we were presented with the challenge of confirming that Actr10 has a specific function in mitochondrial

retrograde transport and reconciling our work with these previous studies. As outlined above, our evidence for a role for Actr10 in mitochondrial transport specifically includes: (1) *actr10* mutant axons are not phenotypically identical to *dync1h1* or *p150* mutants; (2) mitochondrial retrograde, but not anterograde, transport is disrupted in *actr10* mutants while all mitochondrial movement is inhibited in *p150a/b* mutants; and (3) localization and transport of all other cargos are not disrupted in *actr10* mutants. Together, these results argue that loss of Actr10 does not impact all dynein-dynactin function. In addition, a variant of Actr10 lacking the dynactin binding domain retains its ability to interact with mitochondria, further substantiating a direct role for this protein in mitochondria-dynactin interaction. Together, our data argue that Actr10 participates in mitochondrial attachment to the retrograde motor protein complex in neurons in addition to potentially facilitating Arp1 capping and nuclear positioning in other contexts.

## Drp1 and Actr10 function together to regulate mitochondrial retrograde transport

The interaction between Drp1 and Actr10 in mitochondrial retrograde transport may provide a link between mitochondrial movement and mitochondrial fission. Originally, we investigated the role of Drp1 in retrograde mitochondrial transport as previous studies revealed the unique ability of this GTPase to elicit mitochondrial localization to microtubule minus ends (*Smirnova et al., 2001, 1998*). Upon further investigation, it became apparent that a select number of proteins implicated in mitochondrial fission and fusion, including Drp1, have been shown to be essential for mitochondrial transport. Specifically, the fusion-related protein Mitofusin binds to the Miro-Milton complex and is necessary for mitochondrial movement (*Misko et al., 2010*). Similar to Miro and Milton, loss of Mitofusin results in decreased anterograde and retrograde mitochondrial transport in cultured neurons (*Misko et al., 2010*). Interestingly, the inner mitochondrial membrane protein Opa1, also necessary for mitochondrial fusion, does not participate in transport (*Misko et al., 2010*). Thus, eliminating mitochondrial fusion itself does not impact mitochondrial movement; rather mitochondrial outer membrane proteins may have dual roles in fusion-fission dynamics and transport.

Similarly, our work in conjunction with other studies supports the argument that Drp1 participates in both mitochondrial fission and retrograde mitochondrial transport. Drp1 translocates to mitochondria where it can bind to receptors Fis1 and Mff (*LosonLosón et al., 2013*). In its GTP-bound form, it then oligomerizes, forming a constrictable collar necessary for fission during which GTP is exchanged for GDP (*Ingerman et al., 2005*). We found that a Drp1 variant incapable of binding GTP stimulates mitochondrial accumulation at the nuclear periphery, linking fission with transport. Our genetic interaction data suggest that Actr10 functions with Drp1 in this process, though further experiments are necessary to confirm this. Rather than Actr10 working with Drp1 to regulate mitochondrial localization, it is also possible that loss of Actr10 results a decrease in the motile pool of mitochondria in axons upon which Drp1$^{K38A}$ could work. We cannot differentiate between these possibilities at this point. Nevertheless, given the intriguing data generated here, it is tempting to speculate that, upon the GTP-GDP transition, not only do mitochondria undergo fission but Drp1 also recruits the retrograde motor protein complex through interaction with Actr10 to facilitate separation of these daughter mitochondria. The precise nature of Drp1-Actr10 interaction and the role of this interaction in mitochondrial retrograde transport is a subject of current investigation.

## Conclusions

The coordinated regulation of mitochondrial transport is necessary for the formation and maintenance of active neural circuits. Our work identifies Actr10 as a crucial protein for mitochondrial interaction with the dynein-dynactin complex, regulating retrograde transport. In addition, we argue that the role of Actr10 is specific and not due to a general loss of dynactin stability or dynein function. The strongest pieces of evidence supporting this assertion are the ability of Actr10 to bind mitochondria in the absence of dynactin and the disparate nature of the *actr10-dync1h1* and *actr10-p150a/b* mutant phenotypes in axons. In addition, we identified Drp1, a GTPase implicated in mitochondrial fission, as an Actr10 interactor with a potential role in mitochondrial retrograde transport. Together with previous studies on mitochondrial motility, a complex picture is emerging in which mitochondrial transport and fission/fusion are dependent on a core group of proteins, with an independent set of proteins regulating mitochondrial docking. In conclusion, further work is crucial to

understanding how mitochondrial transport and dynamics are orchestrated in axons, which will ultimately allow us to better understand how this critical organelle is positioned and maintained. As mitochondrial transport and dynamics have been implicated in neurological disorders, a more complete mechanistic understanding of their regulation will provide insight into disease pathology.

## Materials and methods

### Zebrafish husbandry and strains

Adult *AB and WIK strains were maintained at 28.5°C and spawned according to standard protocols (*Westerfield, 2000*). ENU mutagenesis and mutant screening were performed as described previously (*Mullins et al., 1994*). Additional strains used include: *TgBAC(neurod:egfp)$^{nl1}$* (*Obholzer et al., 2008*), *actr10$^{nl15}$*, *dync1h1$^{mw20}$* (*Insinna et al., 2010*), *p150a* (also known as *mok$^{s309}$*; (*Del Bene et al., 2008*; *Wehman et al., 2005*), and *p150b$^{nl16}$*.

### RNA-seq based identification of the actr10$^{nl15}$ mutation and genotyping

RNAseq-based mapping was performed according to established protocols (*Miller et al., 2013*). *actr10* heterozygotes (*AB background) were crossed to WIKs to generate a mapping strain. Heterozygous F1 hybrids were identified using pair wise crosses. Eighty wildtypes/heterozygotes and 80 *actr10* mutants were obtained from a single mapping pair. These larvae were lysed at 4 dpf in Trizol according to the manufacturer's protocol. mRNA was extracted in phenol:chloroform:isoamyl alcohol and twice in chloroform before the pellet was redissolved in 20 μL of RNase/DNase free water. Sequencing was performed by the OHSU Massively parallel sequencing core on an Illumina HiSeq2000, generating approximately 30 million reads per condition. From this RNAseq data, RNA-mapper identified a single nucleotide change in the start codon of the *actr10* gene in the *actr10$^{nl15}$* mutant pool. The mutation (validated by sequencing and restriction digest of individual samples) completely segregated with the mutant phenotype. For experiments, *actr10* mutants were identified by genotyping according to the following protocol: PCR amplification of the region around exon 1 of the *actr10* locus resulted in a 374 bp product (forward primer: 5'-CTGTTTTCGGATGAACTGCC TG; reverse primer: 5'-AGATGCTCTTCGTCTTCTGGCTA. The *actr10* mutation inserts a HaeIII cut site. Digestion with HaeIII generates wildtype (209 bp) and *actr10* mutant (195 bp) bands.

### Cloning the zebrafish ortholog of actr10, plasmid production, TALEN synthesis, and mRNA synthesis

The full-length open reading frame of *actr10* was cloned from a 2 dpf zebrafish cDNA library and ligated into the pSC-A-amp/kan vector (Agilent Technologies; forward primer: 5'-AAATGCCCTTG TTTGAA; reverse primer: 5'-TTTCTCAGTGGAGAAGG). All expression and mRNA synthesis vectors were constructed using Gateway compatible cloning as described (*Kwan et al., 2007*). For Actr10 expression, the Actr10 open reading frame was PCR amplified with BP arms and inserted into the Gateway compatible 3'-entry vector pDONRP2R-P3 (forward primer: 5'-ggggacaagtttgtacaaaaaag-caggctAAATGCCCTTGTTTGAA; reverse primer: 5'-ggggaccactttgtacaagaaagctgggtaTTTCTCAG TGGAGAAGG). Other constructs used included: *5kbneurod:mito-TagRFP* (modified from (*Fang et al., 2012*); *5kbneurod:mRFP-dync1li1V2* (*Drerup and Nechiporuk, 2013*); *Hsp701:mRFP-polyA*; *Hsp701:Drp1-mRFP*; *Hsp701:Drp1$^{K38A}$-mRFP*; and *CMV/SP6:EGFP-dync1i2b*. They were constructed as described here. The zebrafish *dync1i2b* ortholog was cloned into the pDONRP2P3 3' entry vector using BP-competent primers (forward primer: 5'-ggggacagctttcttgtacaaagtggAGA TGGCTCCCGTTTTAGAGAAG; reverse primer: 5'-ggggacaactttgtataataaagttgtTCATGCCTCGTTC TCTGTC) and a 2 dpf zebrafish cDNA library. The zebrafish Drp1 ortholog was cloned from 2 dpf cDNA, sequence verified, and cloned into the pDONR221 middle entry vector using BP cloning (forward primer: 5'-GGGGACAAGTTTGTACAAAAAAGCAGGCTTCAACGCGATGGAGGCTCTTA; reverse primer: 5'-GGGGACCACTTTGTACAAGAAAGCTGGGTACCACAAGTGCGTC). To generate *Drp1$^{K38A}$*, the middle entry vector was mutated to change Lysine 38 to Alanine (forward primer: 5'-CGCAGAGTAGCGGGGCGAGTTCAG; reverse primer: 5'-CCAAAACTGAACTCGCCCCGCTAC). All expression plasmids were constructed by LR cloning using the *5kbneurod* (*Mo and Nicolson, 2011*), *CMV/SP6*, or *Hsp701* 5' entry vectors and the pDestTol2pA2 destination vector. mRNA was

synthesized using the SP6 mMessage Machine kit (Life Technologies). TALENs targeting exon 1 of *actr10* were designed using freeware (http://www.talendesign.org) and synthesized as described (*Dahlem et al., 2012*); Tal1 target: AATTTCTGCTGATAAAATGC; Tal2 target: GGGCAGCGGAG-GAGA).

## Generation of the *p150*$^{nl16}$ mutant and genotyping

The *p150b*$^{nl16}$ mutant was generated using CRISPR/Cas9-mediated genome editing as described (*Hwang et al., 2013*; *Shah et al., 2015*). Guide RNA targeting exon 3 (GGCGAGGGCAGCGC TCCCAC) of the *p150b* locus was produced using a PCR-mediated scaffold method as described (*Gagnon et al., 2014*). Cas9 mRNA (*Chang et al., 2013*) was produced using the SP6 mMessage machine kit. Zygotes derived from *p150a* (*moks*$^{309}$) heterozygous outcrosses were injected with 10 pg guide RNA and 100 pg Cas9 mRNA at the one cell stage. Progeny were raised until adulthood and outcrossed to wildtypes to generate larvae. Larvae were genotyped using PCR amplification of the exon 3 locus (forward primer: 5'-CGTGAGTGAGCTCTTGGTCTG; reverse primer: 5'-GCGTCTA TAACCATGTTTGACCTTG) and cut with MwoI to identify insertions/deletions. Changes to the guide RNA target result in loss of the MwoI cut site in *p150b* mutants: wildtype (164 bp + 64 bp) from *p150b* mutants (224 bp).

## *In situ* hybridization and antibody labeling

*In situ* hybridization and antibody labeling were done according to established protocols (*Drerup and Nechiporuk, 2013*). The *actr10* open reading frame was used for in situ hybridization. An ~800 base pair portion the 3'UTR of *p150a* and *p150b* were used for *in situ* hybridization of these paralogues. Antibodies used were: α-Dynein heavy chain (Protein Tech; #12345–1-AP); α-p150 (BD Transduction Laboratories; #610473); α-Lamp1 (Iowa Hybridoma Bank, #1D4B); α-pJNK (Cell Signaling Technology, #9251S); α-GFP (Aves Labs Inc., #GFP-1020); α-DsRed (Clontech, #632496); α-Cytochrome c (BD Biosciences, #556432); α-GFP (Fisher Scientific, #A-11122); α-c-myc (Santa Cruz, #sc-40); and α-ATP$\beta$ (Abcam; #ab128743).

## Cell culture, immunoprecipitation and mitochondrial fractionation

HEK293T cells (Sigma Aldrich; #12022001; identity authenticated by STR-PCR) were cultured according to standard protocols in DMEM with 50 U/mL Pen-Strep. Cells tested mycoplasma free (MycoAlert Mycoplasma Detection Kit; Lonza; L07-218). This human cell line was used for its transfection ease and results were confirmed in vivo. Transfection was done in a 6-well plate using 2.5 μg total DNA and 5 μL of Lipofectamine2000 per well according to manufacturer's protocols (Thermo Fisher; #11668027). The mammalian GFP-Actr10 clone was used previously (*Yeh et al., 2012*) and obtained from Addgene (plasmid #51398). *Drp1-myc* and *Drp1*$^{K38A}$*-myc* expression constructs were previously described (*Smirnova et al., 2001*). GFP-Actr10 deletion constructs were made using the Quik-Change II XL Site-Directed Mutagenesis kit (Agilent, #200521).

Mitochondrial fractionation was performed according to established protocols (*Prudent et al., 2013*). Briefly, for each biological replicate 100 wildtype/heterozygous or *actr10*$^{nl15}$ mutant larvae were disrupted in 700 μL MB buffer (210 mM mannitol, 70 mM sucrose, 1 mM EDTA, 10 mM HEPES pH = 7.5 and protease inhibitors) using a 26 gauge needle. After removal of the input fraction, lysates were subjected to increasing centrifugation speeds to fractionate mitochondria from a heavy and light fractions. The mitochondrial fraction was washed once in fresh MB buffer prior to addition of Laemmli buffer to the pellet and heat denaturation. Extracts were run on a 10% acrylamide gel, transferred to PVDF membrane, and incubated with antibodies prior to developing with the West Pico Substrate (Thermo Fisher; #34080). After initial development, membranes were washed in water and PBS/0.1% Tween prior to incubation with the α-ATP$\beta$ antibody.

## *In vivo* analysis of axonal transport

Axonal transport analyses were done as described (*Drerup and Nechiporuk, 2013*). Briefly, zygotes were injected with plasmids to express cargos of interest tagged with fluorescent proteins. At 4 dpf, larvae (each a biological replicate) were screened for pLL ganglion expression of cargo fusions and imaged using a 63X/NA1.2 water objective on an FV1000 confocal microscope (Olympus). All imaging was done in a single z-plane, allowing acquisition at 2–4 frames per second, as required by the

Nyquist sampling theorem (*Nyquist, 1928*). Kymograph analyses of cargo movement were performed using Metamorph (Molecular Devices).

## Transmission electron microscopy

Zebrafish larvae at 5 dpf were fixed and prepared for TEM analysis using standard protocols (*Czopka and Lyons, 2011*). Samples were embedded and sectioned from the tail of the larvae. An ultramicrotome was used to cut 1000 nm sections, which were then stained with Toluidine Blue to ensure the presence of neuromasts. Subsequently, 70 nm sections were cut and placed on copper mesh grids. Grids were stained with uranyl acetate for one hour and then each grid was cleaned with milliQ water for 45 s. The next day, grids were stained with Sato's lead stain for 15 min and again cleaned with milliQ water. The Washington University Center for Cellular Imaging Jeol JEM-1400 (Jeol USA) electron microscope was used to observe samples and an AMT V601 camera captured images.

## Image and statistical analyses

Volumetric analyses to determine mean fluorescence intensity of immunolabeling was done using Imaris (Bitplane). For this analysis, a surface rendering was done using expression of the *neurod:egfp* transgene in axon terminals and the mean fluorescence intensity was determined. Background from the same channel was subtracted prior to analysis. For quantification of western blots, mean intensity of was determined for signal and background using ImageJ. Statistical analyses were done in JMP. Prior to parametric analyses, data normality and variance were determined. Parametric data was analyzed using ANOVAs with Tukey-Kramer HSD contrasts for multiple comparisons. Non-parametric data was analyzed using Wilcoxon/Kruskal Wallis Tests. Image analyses and figure preparation was done using ImageJ, Adobe Photoshop, and Adobe Illustrator.

## Acknowledgements

We would like to thank members of the Nechiporuk lab for critique of this work, A Forbes and M Culbertson for curating the screen, and C Riso and E Hunt for zebrafish husbandry. A Miller was instrumental in mapping the *actr10$^{nl15}$* mutation using RNAmapper. We are indebted to S Lusk for her work in genotyping and maintaining the mutant strains used here. C De Palma provided the myc-Drp1 constructs. Funding to CMD (NINDS/NIH: K99NS086903), AVN (NICHD/NIH: R01HD072844 and OHSU Center for Spatial Systems Biomedicine: GBMEN0245A1), ALH (F31 NS096814 and Philip and Seema Needleman graduate student fellowship), and KRM (Harry Weaver Scholar of the National Multiple Sclerosis Society) supported this work.

## Additional information

### Funding

| Funder | Grant reference number | Author |
| --- | --- | --- |
| National Institute of Neurological Disorders and Stroke | 1K99NS086903 | Catherine M Drerup |
| Eunice Kennedy Shriver National Institute of Child Health and Human Development | R01HD072844 | Alex V Nechiporuk |
| OHSU Center for Spatial Systems Biomedicine | GBMEN0245A1 | Alex V Nechiporuk |
| National Institute of Neurological Disorders and Stroke | F31 NS096814 | Amy L Herbert |
| Philip and Seema Needleman | Graduate Student Fellowship | Amy L Herbert |
| National Multiple Sclerosis Society | Harry Weaver Scholar | Kelly R Monk |

The funders had no role in study design, data collection and interpretation, or the decision to submit the work for publication.

## Author contributions

CMD, Conceptualization, Data curation, Formal analysis, Funding acquisition, Investigation, Methodology, Writing—original draft, Writing—review and editing; ALH, Formal analysis, Methodology, Writing—review and editing; KRM, Formal analysis, Supervision, Methodology, Writing—review and editing; AVN, Conceptualization, Data curation, Formal analysis, Supervision, Funding acquisition, Investigation, Writing—review and editing

## Author ORCIDs

Catherine M Drerup, http://orcid.org/0000-0002-0219-3075

Alex V Nechiporuk, http://orcid.org/0000-0002-8295-8188

## Ethics

Animal experimentation: This study was performed in strict accordance with the recommendations specified in the Oregon Health and Science University Guide for the Care and Use of Laboratory Animals. All animals were handled in accordance with the institutional animal care and use committee (IACUC) protocol # IS00002972.

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
