## [Decision Letter]

Thank you for submitting your article "Regulation of mitochondria-dynactin interaction and mitochondrial retrograde transport in axons" for consideration by *eLife*. Your article has been reviewed by two peer reviewers, and the evaluation has been overseen by Tanya Whitfield as Reviewing Editor and Jonathan Cooper as the Senior Editor. The reviewers have opted to remain anonymous.

The reviewers have discussed the reviews with one another and the Reviewing Editor has drafted this decision to help you prepare a revised submission.

This interesting study describes a forward screen in zebrafish that identifies a component of the retrograde transport machinery. The authors propose that the Actr10 protein acts as a linker between mitochondria and the dynein/dynactin complex. Specifically, the authors set out to show that in Actr10 mutant zebrafish, retrograde transport of mitochondria, but not other cargo, is specifically compromised. Dynein transport is apparently normal in *actr10* mutants, and mutants in dynein and dynactin components have more severe and widespread phenotypes than *actr10*, supporting the notion that Actr10 functions specifically in transport of mitochondria. The authors suggest that Acrt10 interacts physically with the dynactin complex and brings it to mitochondria. Furthermore, the authors identify a region of the Actr10 protein that is required for physically interacting with the dynactin complex, but not mitochondria. Finally, biochemical and genetic experiments demonstrate that Actr10 interacts with the mitochondrial protein Drp1 and that a putative gain-of-function Drp1 mutant requires Actr10 to increase retrograde transport, supporting the idea that Drp1 recruits Actr10 to mitochondria.

Essential revisions:

1) The results showing that the trafficking effects are specific to mitochondria are not completely convincing, and require further experiments and controls. It is not clear that mitochondria accumulate more than other cargoes in axon swellings. The images should be analysed more quantitatively, avoiding potential effects of swelling size by normalising to area/volume. Anterograde and retrograde traffic should be quantified. (After discussion, it was felt that kymograph analysis would be very suitable here; if there were no comparable deficit in retrograde transport of other cargoes, the conclusion about specificity would be supported further. EM analysis might be useful but is not essential.)

2) Critical controls are absent from several figures. Please see the specific comments from Reviewer 2.

Please see the individual reviews, which are appended below, for further information.

Reviewer #1:

The conclusion about mitochondrial specificity would be even stronger if the authors analyzed more cargo using kymograph analysis-not just lysosomes and JNK. In a previous paper, the authors looked at a wide variety of cargo (Drerup and Nechiporuk, 2013, PLoS Genetics), so this should be technically feasible.

In addition to the kymograph analysis, the authors could quantify enrichment in an alternative, less biased way. For example, perhaps they could outline the entire neuromast and determine the average pixel intensity for each marker over the entire structure in *actr10* and wt. Another good way to do it would be to normalize the intensity of each marker by taking a ratio to a volume-filling reporter, such as cytoplasmic GFP. If they are correct, the cyt10 to GFP ratio should be higher in *actr10* mutants relative to wt, but the other markers should be the same in both genotypes.

One issue that the authors may wish to address a bit more, either through discussion or experiments, is the generalizability of these results. Is mitochondrial transport also affected in other neurons in the Actr10 mutant fish? What is the ultimate consequence of these defects to the neuron? Do they eventually degenerate? What is the ultimate fate of these fish? Is the Actr10 locus associated with any human diseases?

Reviewer #2:

I have raised a few queries into methodological and interpretative issues below. One particular issue is the "mitochondria specific" effect that is not supported by the Lamp1 and pJNK staining. I suggest the authors reconsider this.

Results section and Figure 2. I think it is problematic to say that accumulations don't contain other cargoes. 2I and 2K both show labeling that colocalises with the axonal swelling. Indeed comparing the pJNK and Lamp1 staining in *actr10* embryos (Figure 2) with the same staining in p150a/b (Figure 8) seems to show exactly the same effect (although the opposite conclusion is drawn).

The analysis shown in the graphs in Figure 2 seems to have selected regions outside of swellings for LAMP and pJNK, but inside a swelling for Mitochondria. Is this a strange choice?

In my opinion the swelling doesn't just contain mitochondria. This is true for other genetic mutations that give rise to axonal swelling in the literature.

In relation to the specificity of *actr10* mutation for mitochondria in comparison to pJNK and LAMP it would be necessary to demonstrate that transport of these cargoes is not affected in *actr10* mutant axons. This could be achieved using kymograph analysis.

EM could also be used to identify the contents of these swellings. Alternatively, the authors should remove the claim that these swellings are purely a consequence of mitochondrial accumulation.

Results section and Figure 3. Ideally Figure 3 should show at least one retrograde mitochondrion to allow us to see the rate of retrograde transport. This is potentially important in relation to the point about reversals below.

Results section and Figure 4. To be more certain of the result, the rescue of axonal swelling in a cell specific fashion in neuromasts should be quantified. A single example is given. It would be appropriate to determine the efficiency of rescue in at least 10 neuromasts from 10 embryos.

Results section and Figure 5. This panel shows the swollen axon phenotype, and these swollen axons contain high levels of p150. I would say that this suggests that p150 is mislocalised.

Figure 5. Studying the transport parameters and looking at the kymograph suggests that the effect of *actr10* mutation is to increase the number of reversals of direction in puncta undergoing retrograde movement. Looking at retrograde puncta in I and J, they both show pauses, but in I these are brief and only include small reversals in direction. However in J the retrograde puncta seem to frequently reverse direction. To investigate this it would be appropriate to measure the number of reversals per mito/minute for both anterograde and retrograde cargoes. If correct, this observation would suggest that *actr10* is needed for the "tug of war" between kinesin and dynein motors, and loss of *actr10* leads to more kinesin mediated movement of cargoes that would normally be transported by dynein. This is consistent with the idea that the dynactin complex regulates co-ordinated movement of cargoes in vivo (e.g. Gross JCB 2002).

Figure 5 and O. Immunoprecipitations using "beads only" i.e. omitting the primary antibody are an essential control. GFP is notoriously "sticky" in these types of experiment.

Figure 5. In the co-IP of p150 with exogenous i2b it is critical to include the relevant control. i.e. western blot showing that the p150 signal is only present when i2b is pulled down. (co-IP of uninjected *actr10* extracts performed at the same time as co-IP from i2b injected embryos.) This is in addition to the comment above.

Figure 9. This is an important figure, using biochemistry to address localization of p150 in mutants.

A&B: I am confused by the fact that the heavy fraction is strongly enriched for mitochondria. This suggests incomplete lysis of embryos. Probably just a technical issue, because 9F does not show the same enrichment?

There appears to be less p150 in *actr10* lysate. This in itself might affect transport and needs to be investigated.

The loss of p150 from the mito fraction is striking, but the appearance in the supernatant cannot be correctly interpreted without using non-mitochondrial loading controls. I would suggest probing these blots with the following antibodies to ensure these fractions are "pure". (i) a nuclear protein that should label the heavy fraction (ii) another mito protein to confirm ATPβ. (iii) tubulin (DM1A) to confirm supernatant and mito fractions.

These controls will also confirm equal loading for WT and *actr10*, particularly important in the case of the light fraction.

C-D: control IPs are not presented to show how much p150 is immunoprecipitated using "beads only". This needs to be shown for each experiment. Also the level of GFP immunoprecipitated in Δ7 to Δ13 is much lower, so it is difficult to compare the efficiency of p150 IP.

E: Again controls are missing. Is this specific?

Figure 10. To interpret the K38A result correctly there needs to be a panel showing WCE for GFP-*actr10*. Beads only controls (no primary antibody) are also important here.

Figure 10. I find interpretation of this difficult. It shows that K38A impairs mito transport in WT cells, but since there are already less mitos in the *actr10* axons it is less compelling to say that there is no effect of K38A because of a loss of interaction. i.e. if the effect of mutant *actr10* on mitos is independent of the effect of DRP1 won't we still get the same result?

[Editors' note: further revisions were requested prior to acceptance, as described below.]

Thank you for resubmitting your work entitled "Regulation of mitochondria-dynactin interaction and mitochondrial retrograde transport in axons" for further consideration at *eLife*. Your revised article has been favorably evaluated by Jonathan Cooper (Senior editor), Tanya Whitfield (Reviewing editor), and two reviewers.

The manuscript has been improved but there are some remaining issues that need to be addressed before acceptance, as outlined below (see comments from Reviewer 2). In particular, it is essential that the immunoprecipitation experiments shown in Figure 6 are shown with all the controls. Please add the necessary controls and label the blots as has been done for Figure 11, i.e. indicate which antibodies have been used for the immunoprecipitation and which for the immunoblot.

Reviewer #1:

The authors have been responsive to reviewers and improved the manuscript in several notable ways: First, they have performed electron microscopy to demonstrate that mitochondria accumulate in terminals of *actr10* mutants; second, they have re-analyzed image data with quantification methods that are more convincing and added new measures of transport that support their conclusions; third, they have added critical controls for biochemical experiments; finally, they have revised the text to emphasize alternative interpretations of key conclusions. This is now a thorough and convincing manuscript that identifies a key protein regulating the transport of specific axonal cargoes to maintain healthy axons, making this work of interest to both cell biologists and neuroscientists.

Reviewer #2:

The authors have modified the manuscript, and it shows considerable improvement.

I find 2 areas that require further explanation

1) Reversals:

Results section paragraph four, – Unclear- 0/100 micron in mutants?

Needs to be only in moving mitochondria not per 100 micron, since there are less moving in mutants there is less chance for reversal?

2) IP:

My request: "it is critical to include the relevant control. i.e. western blot showing that the p150 signal is only present when i2b is pulled down. (co-IP of uninjected *actr10* extracts performed at the same time as co-IP from i2b injected embryos.)"

As far as I can tell the new data presented in Figure 6 currently shows no p150 blot data for the additional controls?

---

## [Author Response]

Essential revisions:

1) The results showing that the trafficking effects are specific to mitochondria are not completely convincing, and require further experiments and controls. It is not clear that mitochondria accumulate more than other cargoes in axon swellings. The images should be analysed more quantitatively, avoiding potential effects of swelling size by normalising to area/volume. Anterograde and retrograde traffic should be quantified. (After discussion, it was felt that kymograph analysis would be very suitable here; if there were no comparable deficit in retrograde transport of other cargoes, the conclusion about specificity would be supported further. EM analysis might be useful but is not essential.)

To address this issue, we reanalyzed our immunolabeling experiments to control for axonal volume, assayed the transport of additional cargos in axons, and perform TEM analyses of mutant axon terminals. Altogether, these data demonstrated that there is no identifiable change in the amount of other cargos in *actr10* mutant axon terminals. While we cannot rule out a role for Actr10 in the movement of another cargo, these data indicate that not all cargo rely on Actr10 for retrograde transport.

2) Critical controls are absent from several figures. Please see the specific comments from Reviewer 2.

To address this comment, we repeated a subset of these experiments with bead only controls as recommended. These additional controls confirmed our previous observations and are now included in the revised manuscript. Please find below an outline of our specific responses and where the corrections have been made in the text and figures. Figure 3 and Figure 7 are new while Figure 2, Figure 6 (previously 5), and 11 (previously 9) have had significant changes. Major changes in the text of the manuscript are in blue font.

Please see the individual reviews, which are appended below, for further information.

Reviewer #1:

The conclusion about mitochondrial specificity would be even stronger if the authors analyzed more cargo using kymograph analysis-not just lysosomes and JNK. In a previous paper, the authors looked at a wide variety of cargo (Drerup and Nechiporuk, 2013, PLoS Genetics), so this should be technically feasible.

We performed live imaging experiments to analyze the transport of two additional cargos in response to this review, peroxisomes and late endosomes/lysosomes labeled by Lamp1. JNK transport, as performed in our earlier work, was not feasible at 4 dpf, the time-point at which our other experiments were performed in this paper (our previous JNK analysis was done at 2 dpf). The results of our experiments were very informative. First, we analyzed peroxisome transport. Surprisingly, there were small alterations in the distance of peroxisome retrograde transport and velocity of anterograde peroxisome movement (Figure 7). Upon further literature investigation, it became apparent that peroxisomes and mitochondria actually share many similarities, including membrane proteins. One particular membrane localized protein, Drp1, which is shared by these different organelles, is implicated in Actr10-dependent mitochondrial transport in our work. This hints at a possibility of peroxisome transport being dependent on Actr10-Drp1 interaction (noted in the Discussion section), which will be the focus of our future work. Because peroxisome transport experiments were somewhat difficult to interpret, we analyzed the transport of a third cargo, lysosomes/late endosomes. This analysis demonstrated no change in Lamp-1 labeled lysosome/late endosome transport. Together, these experiments determined that, while peroxisome transport showed a trend towards decreases in retrograde transport frequency (p=0.07; Figure 7), not all cargo transport relies on this Actr10. These additional experiments are described in subsection “Dynein localization and motility does not rely on Actr10” and shown in Figure 7.

In addition to the kymograph analysis, the authors could quantify enrichment in an alternative, less biased way. For example, perhaps they could outline the entire neuromast and determine the average pixel intensity for each marker over the entire structure in actr10 and wt. Another good way to do it would be to normalize the intensity of each marker by taking a ratio to a volume-filling reporter, such as cytoplasmic GFP. If they are correct, the cyt10 to GFP ratio should be higher in actr10 mutants relative to wt, but the other markers should be the same in both genotypes.

We agree this type of analysis is indeed more suitable for signal quantification in axon terminals and thank the reviewer for this suggestion. As suggested by the Reviewer, we used volumetric analyses in Imaris software to determine the mean fluorescent intensity relative to the volume of axon terminals (determined using the cytoplasmic EGFP fill to create a “surface”) for Cytochrome c (mitochondria), Lamp1, and pJNK in Figure 2. This analysis revealed elevated Cytochrome c levels but no change in pJNK or Lamp1 in *actr10* mutants. In addition, we used the same approach to analyze the amounts of Dynein heavy chain and p150 in Figure 6 to demonstrate no change in the localization of these components of the dynein motor complex in *actr10* mutants.

One issue that the authors may wish to address a bit more, either through discussion or experiments, is the generalizability of these results. Is mitochondrial transport also affected in other neurons in the Actr10 mutant fish? What is the ultimate consequence of these defects to the neuron? Do they eventually degenerate? What is the ultimate fate of these fish? Is the Actr10 locus associated with any human diseases?

As Reviewer suggested, we elaborated briefly on the general mutant phenotype in the Results section. Specifically, we stated that the mutant is not homozygous viable and both CNS and PNS axons have these large swellings. Due to the limitations of imaging in other types of neurons (those positioned deeper within the embryo) with the current protocols we have in place, we did not visualize mitochondrial transport in other neuronal subtypes at this time. While we do not discuss it in the manuscript, the axons in the *actr10* mutants do not degenerate prior to larval death. In addition, the *actr10* locus is not associated with any disease states save a short report demonstrating transcription of *actr10* is increased in an in vitro model of prion disease (Brown et al., Hum Mol Genet., 2014).

Reviewer #2:

I have raised a few queries into methodological and interpretative issues below. One particular issue is the "mitochondria specific" effect that is not supported by the Lamp1 and pJNK staining. I suggest the authors reconsider this.

Results section and Figure 2. I think it is problematic to say that accumulations don't contain other cargoes. 2I and 2K both show labeling that colocalises with the axonal swelling. Indeed comparing the pJNK and Lamp1 staining in actr10 embryos (Figure 2) with the same staining in p150a/b (Figure 8) seems to show exactly the same effect (although the opposite conclusion is drawn).

The analysis shown in the graphs in Figure 2 seems to have selected regions outside of swellings for LAMP and pJNK, but inside a swelling for Mitochondria. Is this a strange choice?

In my opinion the swelling doesn't just contain mitochondria. This is true for other genetic mutations that give rise to axonal swelling in the literature.

In relation to the specificity of actr10 mutation for mitochondria in comparison to pJNK and LAMP it would be necessary to demonstrate that transport of these cargoes is not affected in actr10 mutant axons. This could be achieved using kymograph analysis.

*EM could also be used to identify the contents of these swellings. Alternatively, the authors should remove the claim that these swellings are purely a consequence of mitochondrial accumulation.*

The points raised by this reviewer are critical to clarify and address for the submitted manuscript. Importantly, we do not to claim that mitochondria are the only cargo which is abnormally localized or transported in *actr10* mutants. Based on our analyses, we can only say that the other cargos we analyzed are localized and transported normally and dynein is capable of moving normally in the retrograde direction. In addition, no other cargo appears significantly accumulated in *actr10* mutant axon terminals based on our TEM analyses. However, it is possible that movement of other cargo, not analyzed in our studies, is also interrupted in *actr10* mutants. We would like to clarify that our main motivation for the analysis of other cargos was to show that loss of Actr10 does not impact all dynein-mediated cargo transport. This point is now discussed throughout the manuscript.

In addition, to more clearly show that at least some non-mitochondrial cargos are localized normally in *actr10* mutants, we reanalyzed the data using volumetric surface analyses in Imaris as described above. This type of analysis of fluorescence intensity within axon terminals defined by the presence of cytoplasmic EGFP (on the same images analyzed previously) demonstrated that Cytochrome c immunofluorescence is significantly increased in *actr10* mutants while Lamp1 and pJNK are not. This is now shown in Figure 2.

We also analyzed the transport of two additional cargos, peroxisomes and lysosomes, using our established protocols. The results are discussed above in regard to Reviewer 1 point 1 and depicted in Figure 7.

Finally, we collaborated with Amy Herbert and Kelly Monk (now co-authors on the paper) to perform TEM analyses of the PNS axon terminals in *actr10* mutants. These analyses clearly show a preponderance of mitochondria in the axon terminal swellings of lateral line axons. This is shown in Figure 3 of our revised manuscript and discussed in subsection “Mitochondria accumulate in actr10 mutant axon terminals due to failed retrograde transport“. While other cargos may in fact accumulate or move abnormally as well, to date we have no evidence of what that cargo may be. This is now stated in subsection “Actr10s role in mitochondrial retrograde transport“:

“Our data does not preclude a function for Actr10 in the retrograde transport of as of yet unidentified cargo in axons or other roles for Actr10 in cells, which could vary based on a number of factors including cell type and developmental stage.”.

Results section and Figure 3. Ideally Figure 3 should show at least one retrograde mitochondrion to allow us to see the rate of retrograde transport. This is potentially important in relation to the point about reversals below.

We have used a different kymograph to show mitochondrial transport in *actr10* mutants in what is now Figure 4.

Results section and Figure 4. To be more certain of the result, the rescue of axonal swelling in a cell specific fashion in neuromasts should be quantified. A single example is given. It would be appropriate to determine the efficiency of rescue in at least 10 neuromasts from 10 embryos.

We have now specified the number of axon terminals analyzed in wildtype and mutant animals as well as the number of biological replicates in subsection “Actr10 functions autonomously in neurons to regulate axon morphology and mitochondrial localization “of the revised manuscript.

Results section and Figure 5. This panel shows the swollen axon phenotype, and these swollen axons contain high levels of p150. I would say that this suggests that p150 is mislocalised.

We quantified the p150 fluorescence intensity using volumetric analyses in Imaris software. This quantification is shown in Figure 6 of the revised manuscript and demonstrates there is no difference in the level of p150 in *actr10* mutant axon terminals relative to controls.

*Figure 5. Studying the transport parameters and looking at the kymograph suggests that the effect of actr10 mutation is to increase the number of reversals of direction in puncta undergoing retrograde movement. Looking at retrograde puncta in I and J, they both show pauses, but in I these are brief and only include small reversals in direction. However in J the retrograde puncta seem to frequently reverse direction. To investigate this it would be appropriate to measure the number of reversals per mito/minute for both anterograde and retrograde cargoes. If correct, this observation would suggest that actr10 is needed for the "tug of war" between kinesin and dynein motors, and loss of actr10 leads to more kinesin mediated movement of cargoes that would normally be transported by dynein. This is consistent with the idea that the dynactin complex regulates co-ordinated movement of cargoes* in vivo *(e.g. Gross JCB 2002).*

We appreciate the question and think the analysis of reversals adds to the value of the manuscript. To address this issue, we quantified the number of reversals in mitochondrial and dynein movement in the revised manuscript. The results are now reported in the text. We observe no change in reversal frequency of mitochondria or dynein-positive cargos in *actr10* mutants in our imaging experiments. Rather, based on the population analyses shown in Figure 4, it appears that mitochondria which fail to attach to the retrograde motor either move in the anterograde direction or become stationary. This could align well with the tug of war model in so much as failure of mitochondria to attach to cytoplasmic dynein leads to a loss of the ability of this motor to participate in the tug of war and increased anterograde mitochondrial transport.

Figure 5. Immunoprecipitations using "beads only" i.e. omitting the primary antibody are an essential control. GFP is notoriously "sticky" in these types of experiment.

Figure 5. In the co-IP of p150 with exogenous i2b it is critical to include the relevant control. i.e. western blot showing that the p150 signal is only present when i2b is pulled down. (co-IP of uninjected actr10 extracts performed at the same time as co-IP from i2b injected embryos.) This is in addition to the comment above.

The need for bead only controls was duly noted. To address this concern, we repeated the specified immunoprecipitation experiments with no primary (bead only) controls. These experiments confirmed that i2b-GFP can interact with the dynein complex and mRFP-Actr10 but not mRFP-Δ7Actr10 can interact with dynactin. These new Western blots are now shown in Figure 6 and Figure 11.

Figure 9. This is an important figure, using biochemistry to address localization of p150 in mutants.

A&B: I am confused by the fact that the heavy fraction is strongly enriched for mitochondria. This suggests incomplete lysis of embryos. Probably just a technical issue, because 9F does not show the same enrichment?

The loss of p150 from the mito fraction is striking, but the appearance in the supernatant cannot be correctly interpreted without using non-mitochondrial loading controls. I would suggest probing these blots with the following antibodies to ensure these fractions are "pure". (i) a nuclear protein that should label the heavy fraction (ii) another mito protein to confirm ATPβ. (iii) tubulin (DM1A) to confirm supernatant and mito fractions.

These controls will also confirm equal loading for WT and actr10, particularly important in the case of the light fraction.

While the Reviewer raised a valid point, we believe that the observed variations in this experiment are due to the less stringent nature of this particular fractionation (compared to other, more precise methods such as sucrose gradient fractionation). This means that the mitochondrial fraction is only highly enriched for mitochondria relative to the other fractions as described by Prudent et al., 2013. The comparison to the mitochondrial loading control is therefore quite important as it shows that, relative to the amount of mitochondrial protein in this fraction specifically, the amount of p150 precipitated is dramatically reduced. To clarify this point, we have quantified the levels of p150 for each fraction relative to the input as well as mitochondrial loading in each fraction. This quantification is now shown as Figure 11. As the reviewer pointed out, mitochondria are also in the heavy fraction but this is likely due to the viscous nature of this fraction and, as the levels of p150 are also equal. In addition, we pointed out that the “light” fraction contains multiple cellular components.

There appears to be less p150 in actr10 lysate. This in itself might affect transport and needs to be investigated.

The amount of p150 in the *actr10* mutant fraction relative to wildtype siblings has now been quantified. In addition, the levels were normalized to the mitochondrial control (ATPβ) in the corresponding fractions to correct for any variation in loading. We felt it was important to normalize levels of p150 to mitochondria in each fraction, as the question we were addressing in this experiment was mitochondrial attachment to dynactin in the presence or absence of Actr10. This is now shown in Figure 11.

Additionally, the reviewer noted potential reduction in the p150 level in *actr10* mutants, as detected by Western blot (Figure 6). As mentioned above, we have not seen any change p150 levels in axon terminals using whole embryo immunohistochemistry (Figure 6). However, our western blots do show decreases in p150 in the whole embryos extracts relative to wildtype siblings. This is likely due to unequal protein loading: Rather than normalizing to total protein amounts, we chose to use the same number of embryos or larvae for wildtype and *actr10* mutants for each experiment. As the *actr10* mutants are slightly smaller than their wildtype counterparts, this could account for the differences. Our evidence to support this is the consistently lower levels of other proteins (i2b-GFP in Figure 6) in *actr10* mutant extracts. This is now noted in the text in subsection “Dynein localization and motility does not rely on Actr10”.

C-D: control IPs are not presented to show how much p150 is immunoprecipitated using "beads only". This needs to be shown for each experiment. Also the level of GFP immunoprecipitated in Δ7 to Δ13 is much lower, so it is difficult to compare the efficiency of p150 IP.

E: Again controls are missing. Is this specific?

Bead only (no primary antibody) controls are now shown for Figure 11 for IP of endogenous p150 by Actr10 and the Actr10 deletion construct. In addition, we noted in the revised text the variability of the immunoprecipitations from HEK cell extracts and further clarified the necessity of immunoprecipitating endogenous proteins to confirm the importance of the identified region for interaction with dynactin. This is now described in subsection “*Mitochondria fail to attach to the dynein-dynactin complex in actr10 mutants*“.

Figure 10. To interpret the K38A result correctly there needs to be a panel showing WCE for GFP-actr10. Beads only controls (no primary antibody) are also important here.

Because of time limitations, we were not able to repeat the Actr10-Drp1 coimmunoprecipitation with whole cell extracts probed for GFP; however this immunoprecipitation was replicated >3 times with consistent results. This is now stated in the text in subsection “*Drp1 functions with Actr10 in mitochondrial retrograde transport*“.

Figure 10. I find interpretation of this difficult. It shows that K38A impairs mito transport in WT cells, but since there are already less mitos in the actr10 axons it is less compelling to say that there is no effect of K38A because of a loss of interaction. i.e. if the effect of mutant actr10 on mitos is independent of the effect of DRP1 won't we still get the same result?

With regard to Drp1K38A’s impact on mitochondrial localization: we agree with the reviewer that the results of this experiment on their own are not entirely conclusive and should be interpreted with caution. This is now explicitly noted in the Discussion section of the revised manuscript. However, we also argue that there are multiple pieces of evidence to support our argument that Drp1 and Actr10 function together. 1) We show that the K38A mutation in Drp1 strengthens the interaction with Actr10. 2) In addition, work from others (Smirnova et al., 1998; Varadi et al., 2004) demonstrates that expression of Drp1^K38A^ relocalizes mitochondria in a microtubule-dependent fashion to microtubule minus ends, implicating the retrograde motor in this process. 3) Our data also shows that loss of Actr10 causes accumulation of mitochondria in axon terminals due to failed transport while Drp1^K38A^ causes accumulation of mitochondria in the cell body, likely due to increased retrograde movement. In contrast, if Drp1K38A functioned in a pathway parallel to Actr10, we would predict: 1) a further reduction in mitochondrial number in the axon; 2) a rescue (at least partial) in mitochondrial density in the axon terminals; and 3) mitochondrial accumulation in the cell body of *actr10* mutants, none of which are observed. Together, we use these data to argue that Drp1 in its GDP-bound form, is working with Actr10 to regulate mitochondrial localization in axons. This argument is now better articulated in the Discussion section.

We also outlined an addition caveat related to this experiment. It is possible that the motile pool of mitochondria (~50% based on our work) is trapped in *actr10* mutant axon terminals, leading to the reduction in number in the axon capable of being transported. Without a motile pool, Drp1^K38A^ would not be able to decrease the number of mitochondria in the mutant axons to the same degree as in wildtype axons. This is now pointed out in the revised text. Future experiments using live imaging of mitochondrial transport would likely be able to discern between these possibilities.

Unfortunately, these types of experiments are not possible without the generation of additional stable transgenic lines which would take approximately 6 months.

[Editors' note: further revisions were requested prior to acceptance, as described below.]

*The manuscript has been improved but there are some remaining issues that need to be addressed before acceptance, as outlined below (see comments from Reviewer 2). In particular, it is essential that the immunoprecipitation experiments shown in Figure 6 are shown with all the controls. Please add the necessary controls and label the blots as has been done for Figure 11, i.e. indicate which antibodies have been used for the immunoprecipitation and which for the immunoblot.*

In the revised manuscript, we have addressed the three additional concerns raised by reviewer two in response to our resubmission. First, we changed the way we analyzed mitochondrial reversal frequency to make it relative to the number of mitochondria present in the axons. Second, we performed the necessary control experiments for Figure 6 panel O. In addition to wildtype and mutant immunoprecipitation results, we now show uninjected wildtype and bead only wildtype controls as requested. These panels are discussed in the figure legend for Figure 6 and are shown in blue font. Finally, we changed our labeling scheme for the immunoprecipitation experiments in Figure 6 to match those in Figure 11 as requested.